# ACCELERATING TASK GENERALISATION WITH MULTI-LEVEL SKILL HIERARCHIES

**Thomas P. Cannon**
Department of Computer Science
University of Bath
Bath, United Kingdom
`tc2034@bath.ac.uk`

**Özgür Şimşek**
Department of Computer Science
University of Bath
Bath, United Kingdom
`o.simsek@bath.ac.uk`

## ABSTRACT

Developing reinforcement learning agents that can generalise effectively to new tasks is one of the main challenges in AI research. This paper introduces Fracture Cluster Options (FraCOs), a multi-level hierarchical reinforcement learning method designed to improve generalisation performance. FraCOs identifies patterns in agent behaviour and forms temporally-extended actions (options) based on the expected future usefulness of those patterns, enabling rapid adaptation to new tasks. In tabular settings, FraCOs demonstrates effective transfer and improves performance as the depth of the hierarchy increases. In several complex procedurally-generated environments, FraCOs consistently outperforms state-of-the-art deep reinforcement learning algorithms, achieving superior results in both in-distribution and out-of-distribution scenarios.

## 1 INTRODUCTION

A fundamental goal of AI research is to develop agents that can leverage structured prior knowledge, either provided or learned, to act competently in unfamiliar domains (Pateria et al., 2021). This is common behaviour in animals. For example, many newborn mammals, such as foals, can walk shortly after birth due to innate motor patterns; and human infants display instinctive stepping motions when supported (Adolph & Robinson, 2013; Dominici et al., 2011). These innate behaviours, shaped by evolution, act as priors that guide goal-directed actions and enable rapid adaptation.

Humans are believed to organise behaviours into a hierarchy of temporally-extended actions, which helps break complex tasks into simpler, manageable steps (Rosenbloom & Newell, 1986; Laird et al., 1987; Brunskill & Li, 2014). For instance, decision making in humans often involves planning with high-level actions such as "pick up glass" or "drive to college," each of which comprises subtasks such as "reach for glass" or "pull door handle." These subtasks eventually decompose into basic motor movements. Notably, parts of this hierarchy are shared between tasks; for instance, both "pick up glass" and "pull door handle" involve similar gripping movements. Such shared temporally-extended actions, often referred to as *skills*, are reusable behaviours that provide significant benefit when applied across tasks. Specifically, skills enable rapid learning of new tasks beyond those previously experienced by leveraging prior knowledge.

Inspired by this human capability, generating such hierarchical organisation in algorithms could allow artificial agents to adapt quickly to new tasks (Heess et al., 2016). By decomposing complex tasks into reusable, temporally-extended actions, agents can leverage shared behaviours across tasks, enabling faster learning and better generalisation.

Despite advances made in representing, learning, and using behaviour hierarchies, generalising these behaviours across diverse tasks remains a significant challenge for artificial agents (Cobbe et al., 2019). Existing approaches struggle with effectively transferring skills to new environments, limiting their ability to adapt to real-world scenarios (Pateria et al., 2021).

Here we introduce Fracture Cluster Options (FraCOs), a framework for defining, generating and using multi-level hierarchical skills that are designed to maximise an estimate of their future usefulness. We present a rigorous empirical evaluation that shows that FraCOs substantially enhances

out-of-distribution learning. It outperforms three state-of-the-art baselines—Proximal Policy Optimization (PPO) (Schulman et al., 2017), Option Critic with PPO (OC-PPO) (Klissarov et al., 2017) and Phasic Policy Gradient (Cobbe et al., 2021)—in both in-distribution and out-of-distribution learning across several environments from the Procgen benchmark (Cobbe et al., 2020).

## 2 BACKGROUND

**Reinforcement learning** aims to identify how agents should act in their environment to achieve their objectives(Sutton & Barto, 2018). The state of the environment and the action choices of the agent in a given state may be discrete, continuous, or multidimensional. Most reinforcement learning problems are framed as a Markov Decision Process (MDP), defined as a tuple $\langle S, A, P, R, \gamma \rangle$, where $S$ is the set of possible states, $A$ is the set of possible actions, $P$ is the transition probability function with $P(s, a, s')$ indicating the probability of transitioning from state $s$ to state $s'$ after taking action $a$, $R$ is the reward function with $R(s, a, s')$ indicating the expected reward when transitioning from state $s$ to state $s'$ via action $a$, and $\gamma \in [0, 1]$ is the discount factor. A policy $\pi$ maps states to a probability distribution over actions, guiding the agent's behaviour.

At each time step $t \geq 0$, the agent observes the current state $s_t$ and chooses action $a_t$ based on its policy $\pi(s_t)$. The environment then transitions into state $s_{t+1}$ and the agent receives reward $r_{t+1}$. A *trajectory* is the sequence of observations $s_0, a_0, r_1, s_1, a_2, r_3, s_3, ...$ that reflects the agent's interacts with its environment. The agent's objective is to learn a policy that maximises some well defined function of the reward it receives. In this paper, we aim to maximise the cumulative discounted return, $G_t = \sum_{k=0}^{\infty} \gamma^k r_{t+k+1}$.

We define an *environment* as the external system with which the agent interacts, characterized by $\langle S, A, P \rangle$. A *task* in a given environment defines, in addition, a reward function $R$ and discount factor $\gamma$.

**Hierarchical Reinforcement Learning (HRL)** organises the decision making of reinforcement learning agents into multiple levels of abstraction. A widely used approach is the options framework (Sutton et al., 1999). An *option* $z$ is tuple $\langle I_z, \pi_z, \beta_z \rangle$, where $I_z$ is the option's initiation set, describing the set of states where the option can be initiated, $\pi_z$ is the option policy that governs action selection while the option is active, and $\beta : S \to [0, 1]$ is the option's termination condition, expressing the probability of termination at a given state. Options provide a structured way to represent *skills*, which are temporally-abstract actions that are expected to provide benefit when reused across tasks.

**Generalisation in reinforcement learning** refers to the ability of an agent to accumulate rewards in environments, or parts of environments, that it has not been explicitly trained on. Generalisation challenges can be categorised based on the relationship between training and testing distributions, either falling within independent and identically distributed (IID) scenarios or extending to out-of-distribution (OOD) contexts. Additionally, generalisation challenges can be classified by the features of the environment that change, including the state space, observation space, dynamics, and rewards. This classification leads to eight distinct generalisation challenges (Kirk et al., 2021).

In this work, we focus on generalisation performance in OOD tasks where the state space $S$ and reward function $R$ vary, while the action space $A$ and the underlying transition dynamics remain constant. This setup reflects real-world scenarios where an agent, such as a robot, operates under fixed physical principles but encounters diverse tasks.

## 3 RELATED WORK

Policy transfer methods, such as Finn et al. (2017), Grant et al. (2018), Frans et al. (2018), Cobbe et al. (2021), and Mazoure et al. (2022), have shown some success in improving generalisation. However, they often struggle in out-of-distribution (OOD) settings, as they typically rely on task-specific adaptation. In contrast, skill transfer methods learn reusable sub-policies, such as options, that can be flexibly composed, offering a more modular and adaptable framework for transfer learning. By decomposing complex tasks into smaller, reusable components, skill transfer preserves and reuses prior knowledge in novel situations, potentially addressing challenges in OOD generalisation (Konidaris & Barto, 2007; Barreto et al., 2019; Tessler et al., 2017; Mann & Choe, 2013). However,

existing approaches are limited to learning a set of skills that choose directly among primitive actions, lacking the ability to operate in a truly hierarchical manner, being able to choose among both primitive actions and other skills.

A small number of methods have been proposed in the literature for learning *multi-level* hierarchies (Riemer et al., 2018; Evans & Şimşek, 2023; Levy et al., 2019; Fox et al., 2017) but these methods have not addressed skill transfer as a mechanism for accelerating task adaptation and generalisation. Additionally, some of these methods, for instance, those by Levy et al. (2019) and Evans & Şimşek (2023), face difficulties in large, complex problems, for example, when using pixel-based representations. Furthermore, state-based sub-goal methods, including Levy et al. (2019) and Evans & Şimşek (2023), struggle to transfer learned sub-goals to different state spaces and fail to account for variability in action sequences required to reach these sub-goals; for example, "booking a holiday" could involve "calling a travel agent" or "using the internet," each demanding different skills. In contrast, FraCOs avoids creating state-based sub-goals, providing a more flexible framework for transfer across state spaces.

Our work is closely related to Discovery of Deep Options (DDO) by Fox et al. (2017). DDO employs an expectation gradient method to construct a hierarchy *top-down* of options from expert demonstrations. However, DDO does not optimise for generality and it remains unclear how the discovered options perform in unseen tasks. Moreover, the reliance on demonstrations limits the development of increasingly more complex abstractions. In comparison, FraCOs builds *bottom-up*, forming progressively more complex abstractions and selecting options based on their expected *usefulness* in future tasks, directly addressing generalisation challenges.

Also related to our work is Hierarchical Option Critic (HOC) by Riemer et al. (2018). HOC generalises the option-critic framework introduced by Bacon et al. (2017) to a multi-level hierarchy. HOC learns all options simultaneously during training. Both option-critic and HOC suffer from option collapse, where either all options converge to the same behaviour or one particular option is chosen consistently (Harutyunyan et al., 2019). Additionally, the added complexity of option-critic methods can slow learning compared to non-hierarchical approaches such as PPO (Schulman et al., 2017; Zhang & Whiteson, 2019). FraCOs addresses these limitations by naturally preventing option collapse through its option selection process.

## 4 FRACTURE CLUSTER OPTIONS

We hypothesise that identifying reoccurring patterns in an agent's behaviour across successfully completed tasks will improve performance on future tasks that the agent has not yet experienced. Correspondingly, we propose an approach to multi-level skill discovery that consists of three stages: (1) identifying similar patterns in an agent's behaviour across multiple tasks, (2) selecting the most *useful* of these patterns—those considered to be the most likely to appear in trajectories of all possible tasks, and (3) using these identified patterns as a basis for generating options for future use. In this section, we discuss each stage in turn.

### 4.1 IDENTIFYING PATTERNS IN AGENT BEHAVIOUR

Our objective is to identify and cluster the most *useful* patterns in agent behaviour that lead to *successful* outcomes. To achieve this, we require a method for capturing and analysing patterns produced during task execution. To this end, we introduce the concept of a *fracture*, defined as a sequence of actions that start at a specific state. A fracture $\phi$ is a tuple $(s, a_1, a_2, \ldots, a_k)$, where $s$ is the start state and $a_1, a_2, ..., a_k$ is a sequence of $k$ actions that can be initiated at state $s$.

Given a set of trajectories $\mathcal{T}$ from tasks that an agent has experienced, we can generate a set of candidate fractures. For a single trajectory $\tau \in \mathcal{T}$ of length $n$, let $\mathcal{F}$ denote the set of all fractures that can be derived from this trajectory:

$$\mathcal{F} = \{(s_t, a_t, a_{t+1}, \ldots, a_{t+b-1}) \mid 0 \leq t \leq n - b\}, \tag{1}$$

where the parameter $b$ controls the temporal length of the fracture. We can repeat this for all $\tau \in \mathcal{T}$ such that $\Phi = \{\mathcal{F}_1, \mathcal{F}_2, \ldots, \mathcal{F}_{|\mathcal{T}|}\}$ is the complete set of fractures derived from all trajectories in $\mathcal{T}$.

**Can fractures capture the underlying structure of the environment?** We explore this question in Four Rooms, a classic grid-based reinforcement learning environment, consisting of four rooms, connected to each other with narrow doorways. The environment has a single goal state; the agent receives a positive reward upon reaching this state, which terminates the episode. This environment is depicted in the top left corner in Figure 1. Further details can be found in Appendix A.7. In all of our grid-world implementations, the agent can observe only a $7 \times 7$ area centred on itself and a scalar indicating the direction of the reward. This is similar to MiniGrid (Chevalier-Boisvert et al., 2024), except that our observations are ego-centric.

We train tabular Q-learning agents in multiple versions of the environment with different goal states, generating trajectories for both the trained agents and an agent selecting actions randomly. We then create fractures following Equation 1, with $b = 2$. To reveal the structural differences between the fractures derived from the random agent and the trained agent, we use UMAP (McInnes et al., 2018), a dimensionality reduction technique that projects the high-dimensional fracture data into two dimensions. UMAP is particularly useful for this task because it preserves local similarities within the data. We show the resulting two-dimensional visualisation in Figure 1. The figure reveals a near-uniform distribution for the agent acting randomly, while the fractures from trained agents form distinct clusters, reflecting the underlying structures in successful trajectories. This observation is consistent across other environments, as shown in Appendix A.7.4.

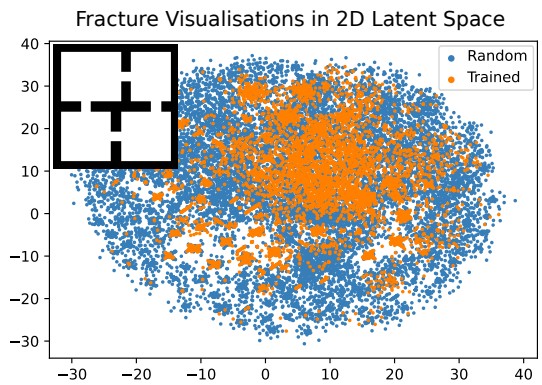

Figure 1: A two-dimensional representation of the fractures ($b = 2$) derived from agents acting for 10,000 time-steps in Four Rooms.

To identify clusters of fractures, we employ unsupervised clustering techniques. Specifically, we use HDBSCAN (Campello et al., 2013) for all tabular methods in this work. Figure 2 shows four clusters randomly selected for visualisation, after fractures with a chain length of $b = 4$ are grouped into clusters using HDBSCAN. For each of the four clusters, the visualisation shows *all* fractures within that cluster, demonstrating that, despite differences in starting states, action sequences, and final states, the fractures within each cluster share similar semantic meanings.

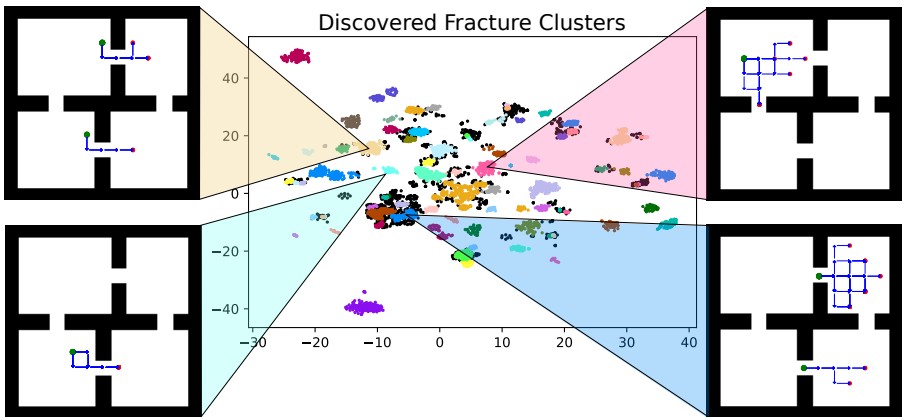

Figure 2: Four examples of discovered fracture clusters ($b = 4$) from agents trained in Four Rooms. Each cluster is represented by a colour. In the four examples, the green circles represent possible starting states, blue arrows indicate actions, the width of the arrows shows the frequency of the state-action pair within the cluster, and the red circles indicate the final states of the fracture.

## 4.2 SELECTING USEFUL FRACTURE CLUSTERS

Fracture clusters formed in Section 4.1 identify behaviours that are similar to each other; however, the number of clusters identified can be very high, with some being highly task specific. Developing all clusters into options may burden the agent with action choices that are not particularly useful. It is therefore essential to identify the clusters that are most likely to be useful in future tasks.

Consider the hypothetical scenario in which we can observe all possible trajectories across all possible tasks. We will consider a trajectory to be successful if its cumulative return exceeds a predefined threshold, similar to the criterion proposed by Chollet (2019)—Table 10 shows all minimum returns used in our work. We use $\mathcal{X}_w$ to denote the set of all tasks with successful outcomes. In this ideal setting, the set of all successful fractures, denoted by $\Phi_w$, can be derived from the set of all successful trajectories, denoted by $\mathcal{T}_w$.

To sensibly select fracture clusters, we must evaluate their potential for reuse in future tasks. We do this by defining the *usefulness U* of a fracture cluster $\tilde{\phi}$ based on its likelihood of contributing to success across tasks. Specifically, usefulness is determined by three factors, described below.

> **Appearance probability,** $P_a(\tilde{\phi})$. For a given fracture cluster $\tilde{\phi}$, this is the probability that a randomly selected successful trajectory $\tau_w \in \mathcal{T}_w$, from a randomly selected successful task $x_w \in \mathcal{X}_w$, will contain at least one fracture $\phi \in \tilde{\phi}$.
>
> **Relative frequency,** $P_f(\tilde{\phi})$. For a given fracture cluster $\tilde{\phi}$, this is the proportion of times that some fracture $\phi \in \tilde{\phi}$ appears among all successful fractures $\Phi_w$.
>
> **Entropy of usage,** $H(\tilde{\phi})$. For a given fracture cluster $\tilde{\phi}$, this is the entropy of at least one $\phi \in \tilde{\phi}$ appearing across all successful trajectories $\mathcal{T}_w$.

The usefulness of a fracture cluster $\tilde{\phi}$ is defined as the mean of these three factors:

$$U_{\tilde{\phi}} = \frac{1}{3}[P_a(\tilde{\phi}) + P_f(\tilde{\phi}) + H(\tilde{\phi})] \qquad (2)$$

Appendix A.17 includes an ablation study that explores the impact of each term.

In an ideal scenario, we could observe all possible tasks and trajectories and directly calculate the usefulness $U_{\tilde{\phi}}$ of each fracture cluster. This is generally not possible. Instead, we must rely on available data, using the tasks and trajectories encountered during training. Let $n$ index individual tasks and $N$ be the total number of experienced tasks. We form $\Phi_w$ from the $N$ experienced tasks.

**Estimating appearance probability.** We approximate $P_a(\tilde{\phi})$ by using a Bayesian approach, modelling the probability that at least one fracture $\phi \in \tilde{\phi}$ appears in a successful trajectory as a binomial likelihood with a Beta conjugate prior. The prior parameters $\alpha$ and $\beta$, both set to 1, reflect an uninformative prior. We define the appearance indicator $\zeta_n$ to be equal to 1 if any fracture $\phi \in \tilde{\phi}$ appears in trajectory $\tau_n$, and 0 otherwise.

**Estimating relative frequency.** We estimate $P_f(\tilde{\phi})$ by counting the occurrences of $\phi \in \tilde{\phi}$ in $\Phi_w$, which we denote as $\mathrm{count}(\tilde{\phi}, \Phi_w)$, and dividing this count by the total number of fractures in $\Phi_w$.

**Estimating entropy.** The entropy $H(\tilde{\phi})$ is approximated using Shannon's entropy formulation (Shannon, 1948). Specifically, the term $\frac{\mathrm{count}(\tilde{\phi}, \tau_w)}{|\tau_w|}$ represents the empirical probability of observing some fracture $\phi \in \tilde{\phi}$ in a trajectory $\tau_w$. This probability is used to compute the entropy. The normalisation factor $N_{\tilde{\phi}}$ ensures that the entropy is scaled appropriately relative to the total number of unique fracture clusters $\tilde{\phi}$.

We derive the full approximation in Appendix A.2. The result is expressed as the *expected usefulness* shown below:

$$E[U_{\tilde{\phi}}] = \frac{1}{3} \left( \underbrace{\frac{\sum_{n=1}^{N} \zeta_n + \alpha}{N + \alpha + \beta}}_{\text{Appearance Probability}} + \underbrace{\frac{\mathrm{count}(\tilde{\phi}, \Phi_w)}{|\Phi_w|}}_{\text{Relative Frequency}} - \underbrace{\sum_{\tau_w \in T_w} \frac{\mathrm{count}(\tilde{\phi}, \tau_w)}{|\tau_w|} \log_{N_{\tilde{\phi}}} \left( \frac{\mathrm{count}(\tilde{\phi}, \tau_w)}{|\tau_w|} \right)}_{\text{Estimated Entropy}} \right) \qquad (3)$$

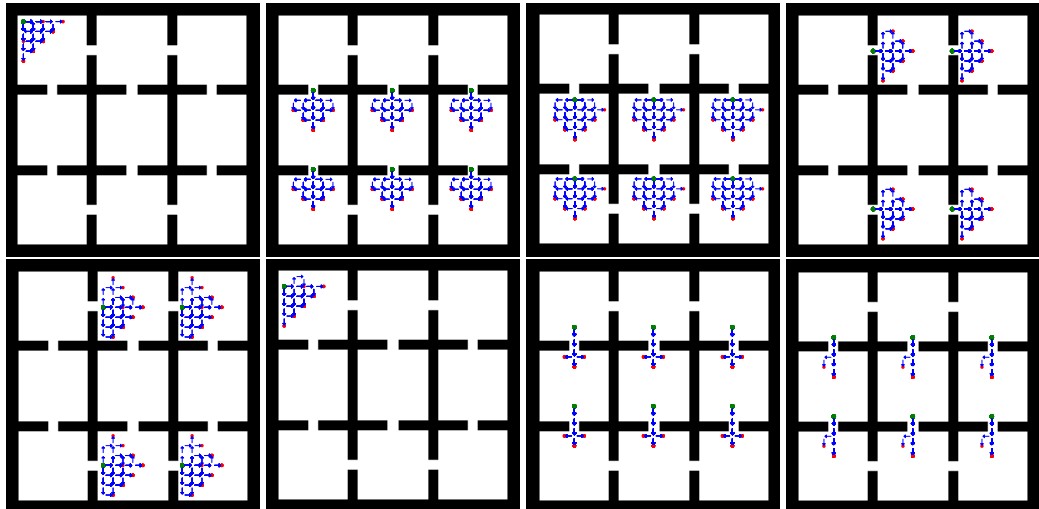

Figure 3: The eight fracture clusters with the highest expected usefulness in Nine Rooms. Expected usefulness decreases from left to right in the top row, then from left to right in the bottom row. Green points represent possible starting states, blue arrows indicate actions, the width of the arrows shows the frequency of the state-action pair within the cluster, and the red points indicate the final states of the corresponding fracture.

Fracture clusters that yield the highest expected usefulness will be selected to be developed into options. To evaluate the effectiveness of Equation 3, we train an agent to reach 100 randomly selected goal states in the Nine Rooms environment, shown in Figure 3. We then collect evaluation trajectories, form fractures with a chain length of $b = 4$, cluster the fractures, and compute their expected usefulness. In Figure 3, we plot the eight fracture clusters with the highest expected usefulness. The fracture cluster with the highest expected usefulness moves the agent from the starting state in all sensible directions, without repetitions of movements. The majority of the other fracture clusters transverse bottlenecks.

**Forming multiple levels of the skill hierarchy.** Once the most useful fracture clusters have been identified, they can be converted into options, extending the agent's action space; this is detailed in Section 4.3. When learning a new task, the agent can now choose from both primitive actions and these higher-level behaviours. To build additional levels of the hierarchy, the process of identifying and clustering fractures is repeated. In subsequent iterations, trajectories—and consequentially, fractures—may consist of a mix of primitive actions and higher-level options. This iterative approach naturally leads to the creation of a multi-level hierarchical structure, where each level captures increasingly complex temporally-extended behaviours.

### 4.3 USING FRACTURE CLUSTERS

To leverage the most useful fracture clusters, we transform them into options we call *Fracture Cluster Options*, or *FraCOs*. A FraCO $z$ is characterised by an initiation set $I_z$, a termination condition $\beta_z$, and a policy $\pi_z$.

**Initiation set.** The initiation set $I_z$ defines the states in which the FraCO $z$ can be initiated. To construct $I_z$, we first consider all possible action sequences of length $b$ (the chain length) denoted as $\mathbf{a} = (a_1, a_2, \ldots, a_b)$, where $a_i \in A, i = 1, ..., b$. We can now define the set of all possible fractures in state $s$ as follows:

$$F_s = \{(s, \mathbf{a}) \mid \mathbf{a} \in A^b\}, \tag{4}$$

where $A^b$ represents the set of all action sequences of length $b$.

For each fracture $\phi \in F_s$, we can estimate the probability that it belongs to a FraCO $z$. The set of fractures assigned to cluster $z$ in state $s$ can be defined as:

$$G_{z,s} = \{\phi \in F_s \mid P(\phi \in z) > \theta\},$$

where $P(\phi \in z)$ denotes the estimated probability that fracture $\phi$ belongs to cluster $z$, and $\theta$ is a threshold parameter. The method for estimating $P(\phi \in z)$ can vary depending on the implementation. In our tabular implementation, we directly use the prediction function provided by HDBSCAN. In our deep implementation, a neural network is trained to predict this probability. A FraCO $z$ can be initiated in state $s$ if $G_{z,s}$ is not empty:

$$I_z = \{s \in S \mid G_{z,s} \neq \emptyset\} \tag{5}$$

**Policy.** When FraCO $z$ is initiated in state $s$, it follows the policy $\pi_z$, as described in Algorithm 1. The policy selects the fracture $\phi_z = (s, \mathbf{a})$ from $G_{z,s}$ that has the highest probability $P(\phi_z \in z)$ and then executes the sequence of actions $\mathbf{a} = (a_1, a_2, \ldots, a_b)$. If one of the selected actions is another FraCO $z'$, the agent must compute $G_{z',s}$ and recursively call the policy until the option terminates.

**Termination condition.** The FraCO $z$ terminates under two conditions: when all actions in the selected fracture $\phi_z$ have been executed or when the agent attempts to execute a nested FraCO $z'$ but cannot find a matching fracture ($G_{z',s} = \emptyset$). The termination condition $\beta_z(s)$ is therefore defined as follows:

---

**Algorithm 1:** Option policy $\pi_z(G_{z,s}, Z, s)$

**Input:** $G_{z,s}$, set of options $Z$, state $s$
**Initialize:** Select the fracture $\phi_z = (s, \mathbf{a})$ from $G_{z,s}$ with the highest probability $P(\phi_z \in z)$, where $\mathbf{a}$ is a sequence of actions $(a_1, a_2, \ldots, a_b)$
**for** *each action $a_j$ in* $\mathbf{a}$ **do**
   **if** *$a_j$ is a primitive action* **then**
     | Execute $a_j$, resulting in new state $s$;
   **else**
     $a_j$ is another option $z'$;
     Compute $G_{z',s}$;
     Recursively call $\pi_{z'}(G_{z',s}, Z, s)$;
   **end**
**end**

---

$$\beta_z(s) = \begin{cases} 1 & \text{if all actions in } \phi_z \text{ have been executed} \\ 1 & \text{if the next action is } z' \text{ and } G_{z',s} = \emptyset \\ 0 & \text{otherwise} \end{cases} \tag{6}$$

**Learning with FraCOs.** FraCOs are fixed once created. The agent learns to choose between primitive actions and available FraCOs using standard reinforcement learning algorithms (e.g., Q-learning for tabular settings, PPO for deep learning implementations).

---

**Example FraCO usage.** An agent in state $s$ considers all possible fractures in that state, $F_s$. For each fracture $\phi \in F_s$, the agent then estimates the probability $P(\phi \in z_i)$ that the fracture $\phi$ belongs to $z_i$, for each FraCO $z_i \in Z$. If a fracture has a probability above the threshold $\theta$, then $z_i$ becomes available as an option.

The agent's policy $\pi$ selects from among the set of available FraCOs and primitive actions. Suppose it selects FraCO $z_1$. The agent picks the fracture $\phi_{z_1}$ with the highest probability $P(\phi_{z_1} \in z_1)$ and executes its action sequence.

As the agent executes these actions, if one of these actions is another FraCO, say $z_2$, then the agent forms a new $F_{s'}$ for the new current state $s'$, estimates $P(\phi \in z_2)$, and selects the most probable fracture for $z_2$, which is above the threshold $\theta$. If the agent cannot find a fracture such that $P(\phi \in z_2) > \theta$, then the option terminates.

This process repeats until the termination condition $\beta_{z_1} = 1$ is met, completing the execution of FraCO $z_1$.

---

## 5 EXPERIMENTAL RESULTS

We evaluate FraCOs in three different experiments. The first experiment focuses on OOD reward generalisation tasks using a tabular FraCOs agent in the Four Rooms, Nine Rooms, and Ramesh Maze (Ramesh et al., 2019) grid-world environments. The second experiment examines OOD state generalisation tasks within a novel environment called MetaGrid, explained in detail in Appendix A.7. The final experiment evaluates a deep FraCOs agent, implemented with a three-level hierarchy using PPO in the procgen suite of environments (Cobbe et al., 2020).

Interquartile Mean Episodic Returns of Different Depths of FraCOs

Figure 4: Episodic returns with a tabular FraCOs agent trained in the Four Rooms, Nine Rooms, and Ramesh Maze environments. Results show interquartile means of 10 independently seeded experiments, with shaded areas indicating the standard error.

We compare FraCOs' performance with CleanRL's Procgen PPO and PPG implementations (Huang et al., 2022) and Option Critic with PPO (OC-PPO) (Klissarov et al., 2017). Further details on baseline implementations are provided in appendices A.11 and A.13.

In the grid-world environments, the agent receives a reward of $+1$ for reaching the goal and an additional $-0.001$ at each time step; episodes have a maximum length of 500 time steps; the fracture chain length is set at $b = 2$. In Procgen environments, we set $b$ to 3. Further details on chain-length selection are provided in Appendix A.8.

## 5.1 EXPERIMENT 1: TABULAR REWARD GENERALISATION

In this experiment, we evaluate FraCOs in the Four Rooms, Nine Rooms, and Ramesh Maze environments, using a fixed state space $S$ while varying the reward function $R$. During the **FraCOs discovery phase**, a tabular Q-learning agent is trained on 50 distinct tasks, each with a unique reward location. After completing training for each task, the top 20 FraCOs are extracted from the agent's final trajectories, as described in Section 4, and incorporated into the action space. This process is repeated iteratively, corresponding to each level of the hierarchy, with FraCOs being added incrementally at each stage.

In the **evaluation phase**, we create four new agents, each with access to a progressively deeper hierarchy of FraCOs. The agents are trained on 10 unseen test tasks—each with a unique reward location—and evaluated periodically. As shown in Figure 4, results demonstrate that learning is progressively accelerated on unseen tasks as the depth of the hierarchy increases.

## 5.2 EXPERIMENT 2: TABULAR STATE GENERALISATION

In this experiment, we evaluate FraCOs in a novel environment called MetaGrid, designed to test state generalisation. MetaGrid is a navigational grid-world constructed from structured $7 \times 7$ building blocks, which can be combined in various ways to create novel state spaces while preserving areas of local structure. The agent observes the environment through a $7 \times 7$ window, consistent with our other grid-world environments. For more details on MetaGrid, refer to Appendix A.7.

During the **FraCOs discovery phase**, a tabular Q-learning agent is trained at each hierarchy level on 100 randomly generated $14 \times 14$ MetaGrid tasks. A task in this experiment corresponds to a unique state space $S$ and reward location $R$. After training, 20 FraCOs are extracted from final trajectories and incorporated into the action space.

In the **evaluation phase**, for each hierarchy level, a separate agent is created. These agents are then trained in two settings: (1) previously unseen $14 \times 14$ domains and (2) larger $21 \times 21$ domains. Periodic evaluation episodes are conducted during training to track performance. As shown in Figure 5, these results also demonstrate that the rate of learning increases with deeper hierarchies.

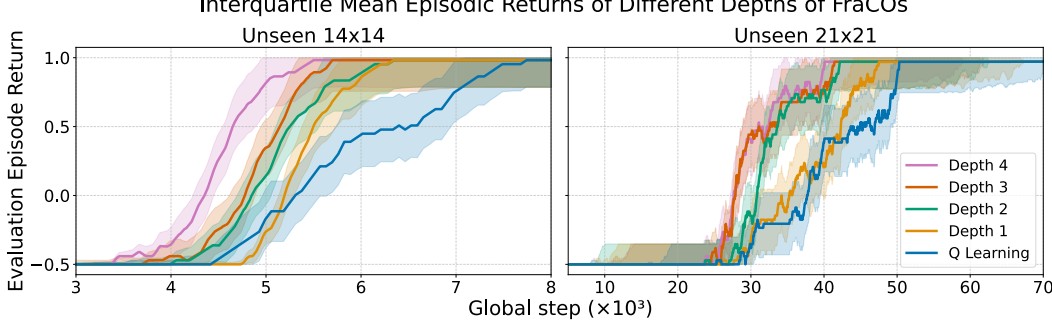

Figure 5: Episodic returns for tabular FraCOs in unseen MetaGrid domains of varying sizes. Results are the interquartile means of 10 independently seeded experiments, with shaded areas indicating the standard error.

## 5.3 EXPERIMENT 3: DEEP STATE AND REWARD GENERALISATION

In this experiment, we test FraCOs in OOD tasks from the Procgen benchmark (Cobbe et al., 2020), which is a suite of procedurally generated arcade-style environments designed to assess generalisation across diverse tasks; further details are provided in Appendix A.7. We compare FraCOs with three methods, Option Critic with PPO (OC-PPO) (Klissarov et al., 2017), PPO (Schulman et al., 2017), and Phasic Policy Gradient (PPG) (Cobbe et al., 2021), across eight Procgen environments, where each task has a unique unseen state space $S$ and reward function $R$.

**FraCOs modifications.** To handle the challenges of applying traditional clustering to high-dimensional pixel data, we simplify the approach by grouping fractures with the same action sequences, regardless of state differences. Additionally, a neural network is used to estimate initiation states and policies, which reduces the computational burden of performing a discrete search over the complex $64 \times 64 \times 3$ state space and managing 15 possible actions during millions of training steps. These modifications do not change the fundamental approach of FraCOs, only its implementation. Full details of these modifications are provided in Appendix A.10, with further information on the experiments, baselines, and hyperparameters in Appendix A.11.

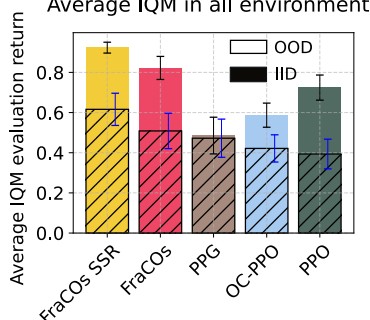

Figure 7: Mean min-max normalised IQM returns with standard errors across Procgen environments.

FraCOs and OC-PPO both learn options during a 20-million time-step warm-up phase, with tasks drawn from the first 100 levels of each Procgen environment. FraCOs learns two sets of 25 options, corresponding to different hierarchy levels, while OC-PPO learns a total of 25 options. After the warm-up, the policy over options is reset, and training continues for an

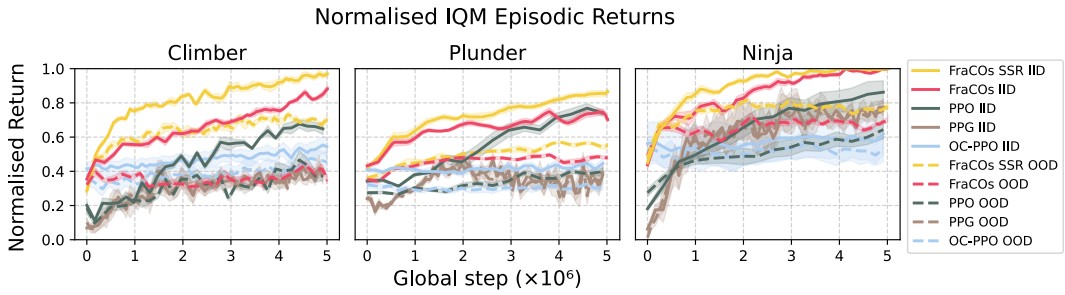

Figure 6: A comparison of learning curves of a sample of three Procgen environments.

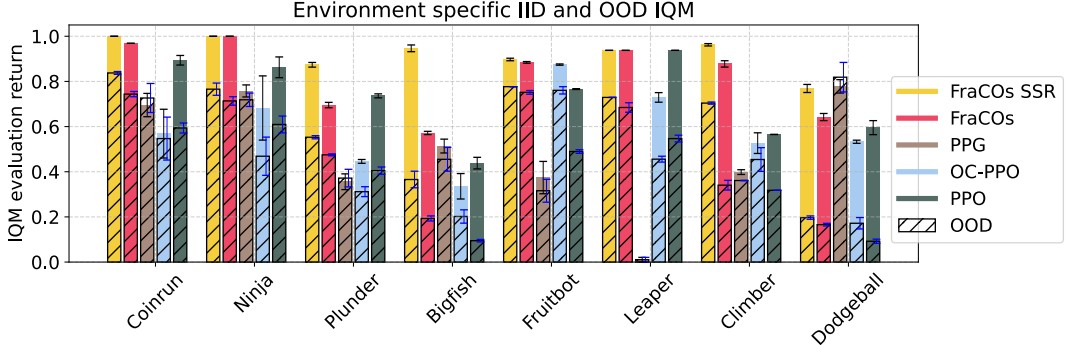

Figure 8: Min-max normalised IQM returns with standard errors in individual Procgen environments.

additional 5 million time steps. During this phase, we periodically conduct evaluation episodes on both IID and OOD tasks, with OOD tasks drawn from Procgen levels beyond 100.

We test two versions of FraCOs: one where the policy over options is completely reinitialised after the warm-up phase, and another that transfers a Shared State Representation (SSR), referred to as FraCOs-SSR. In the SSR, a shared convolutional layer encodes the state, followed by distinct linear layers for the critic, policy over options, and option policies. These convolutional layers are not reset after warm-up, enabling a shared feature representation across tasks. Since OC-PPO inherently relies on a shared state representation to define its options and meta-policy, comparing it with FraCOs-SSR offers a fair evaluation. Without this shared representation, OC-PPO struggles to maintain stable and meaningful options across tasks. Despite this adjustment for fairness, we find that FraCOs, even without SSR, consistently outperforms all baselines. For further implementation details of FraCOs-SSR, refer to Appendix A.11.

Figure 7 provides the mean min-max normalised interquartile mean (IQM) across all Procgen environments. Figure 8 provides the results in each environment. We also provide three sample learning curves in Figure 6. Each experiment was repeated with eight seeds. On average, we observe that FraCOs and FraCOs-SSR are able to improve both IID and OOD returns over all baselines.

## 6 DISCUSSION AND LIMITATIONS

We introduced Fracture Cluster Options (FraCOs) as a novel framework for multi-level hierarchical reinforcement learning. In tabular settings, FraCOs demonstrated accelerated learning on unseen tasks, with performance improving as the depth of the skill hierarchy increased. In deep reinforcement learning experiments, FraCOs outperformed state-of-the-art algorithms OC-PPO, PPO, and PPG on both in-distribution (IID) and out-of-distribution (OOD) tasks, showcasing its potential for robust generalisation.

In its current form, FraCOs has a number of limitations. Clustering methods, such as HDBSCAN, struggle to accurately predict cluster assignments for new data points as environments grow more complex. Simplified techniques and neural networks were introduced here to address this difficulty but further research into scalable clustering solutions is needed. Additionally, this work focused on discrete action spaces; extending FraCOs to continuous action spaces remains as future work. Despite these limitations, FraCOs provides a strong foundation for advancing hierarchical reinforcement learning and improving generalisation across tasks.

### ACKNOWLEDGEMENTS

This work was supported by the UKRI Centre for Doctoral Training in Accountable, Responsible and Transparent AI (ART-AI) [EP/S023437/1] and the University of Bath. We would like to thank the members of the Bath Reinforcement Learning Laboratory for their constructive feedback. This research made use of Hex, the GPU Cloud in the Department of Computer Science at the University of Bath.

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

# A  APPENDIX

## A.1  GLOSSARY OF TERMS AND DERIVATIONS

| Term/Symbol | Definition/Derivation |
|---|---|
| **MDP (Markov Decision Process)** | A mathematical framework for modelling decision-making, defined by the tuple $\langle S, A, P, R, \gamma \rangle$, where: <ul><li>$S$: Set of possible states.</li><li>$A$: Set of possible actions.</li><li>$P$: Transition probability function, $P(s, a, s')$.</li><li>$R$: Reward function, $R(s, a, s')$.</li><li>$\gamma$: Discount factor, $\gamma \in [0, 1]$.</li></ul> |
| $S$ | Set of possible states in an MDP. |
| $A$ | Set of possible actions in an MDP. |
| $P$ | Transition probability function; $P(s, a, s')$ gives the probability of transitioning from state $s$ to $s'$ after action $a$. |
| $R$ | Reward function; $R(s, a, s')$ is the reward received when transitioning from $s$ to $s'$ via action $a$. |
| $\gamma$ | Discount factor in an MDP, $\gamma \in [0, 1]$, representing the importance of future rewards. |
| $s_t$ | State of the agent at time step $t$. |
| $a_t$ | Action taken by the agent at time step $t$. |
| $s'$ | Next state after taking action $a_t$ from state $s_t$. |
| $\pi(s_t)$ | Policy of the agent, mapping state $s_t$ to a probability distribution over actions. |
| $G_t$ | Cumulative discounted return from time $t$, defined as $G_t = \sum_{k=0}^{\infty} \gamma^k r_{t+k+1}$. |
| **Trajectory** ($\tau \in T$) | A sequence of states, actions, and rewards experienced by the agent: $\tau = (s_0, a_0, r_1, s_1, a_1, r_2, \dots)$. |
| **Task** | A task is defined as a unique MDP. |
| **Reward generalisation tasks** | MDPs with $R$ values outside of the training distribution, while $S$, $A$, and $P$, remain within distribution. |
| **State generalisation tasks** | MDPs with $S$ and $R$ values outside of the training distribution, while $A$, and $P$, remain within distribution. |
| **Option** | In HRL, a temporally extended action, defined by: <ul><li>$I$: Initiation set.</li><li>$\pi_z$: Intra-option policy.</li><li>$\beta(s)$: Termination condition.</li></ul> |
| $I$ | Initiation set of an option; the set of states where the option can be initiated. |
| $\pi_z$ | Intra-option policy; the policy followed while the option is active. |
| $\beta(s)$ | Termination condition of an option; gives the probability of terminating the option in state $s$. |
| **FraCOs (Fracture Cluster Options)** | The proposed method for defining, forming, and utilizing multi-level hierarchical options based on expected future usefulness. |
| $\phi$ | A fracture; a state paired with a sequence of actions: $$\phi = (s_t, a_t, a_{t+1}, \dots, a_{t+b-1})$$ |
| | *Continued on next page...* |

| Term/Symbol | Definition/Derivation |
|---|---|
| $b$ | Chain length of a fracture; specifies the number of actions following the state $s_t$. |
| $F$ | Set of fractures derived from a single trajectory: $$F = \{(s_t, a_t, a_{t+1}, \ldots, a_{t+b-1}) \mid 0 \leq t \leq n - b\}$$ |
| $\Phi$ | Complete set of fractures from all trajectories: $$\Phi = \{F_1, F_2, \ldots, F_{|\mathcal{T}|}\}$$ |
| $\tilde{\phi}$ | A fracture cluster; a group of fractures with similar behaviours. |
| **Usefulness Metric** $(U(\tilde{\phi}))$ | A measure to evaluate fracture clusters based on their potential for reuse in future tasks: $$U_{(\tilde{\phi})} = \frac{1}{3} \left( P[\tilde{\phi} \in \tau_w \mid x_w] + P[\tilde{\phi} \mid \Phi_w] + H(\tilde{\phi} \mid \mathcal{X}_w) \right)$$ |
| **Expected Usefulness terms** | <ul><li>$\zeta_n$: Appearance indicator; $\zeta_n = 1$ if any fracture $\phi \in \tilde{\phi}$ appears in trajectory $\tau_n$, else 0.</li><li>$\alpha$, $\beta$: Parameters of the Beta prior distribution, typically set to 1.</li><li>$N$: Total number of experienced tasks.</li><li>$\Phi_w$: Set of all successful fractures.</li><li>$T_w$: Set of successful trajectories.</li><li>$N_{\tilde{\phi}}$: Normalization constant for entropy calculation.</li></ul> |
| **Derivation of Appearance Probability** | Using Bayesian inference, the appearance probability is estimated as: $$P[\tilde{\phi} \in \tau_w \mid x_w] = \frac{\sum_{n=1}^{N} \zeta_n + \alpha}{N + \alpha + \beta} \qquad (7)$$ Where $\zeta_n$ are observations modeled as Bernoulli random variables with a Beta prior. |
| **Derivation of Relative Frequency** | Calculated as: $$P[\tilde{\phi} \mid \Phi_w] = \frac{\text{count}(\tilde{\phi}, \Phi_w)}{|\Phi_w|} \qquad (8)$$ Where $\text{count}(\tilde{\phi}, \Phi_w)$ is the number of times fractures in $\tilde{\phi}$ appear among all successful fractures $\Phi_w$. |
| **Derivation of Entropy of Usage** | The entropy of a fracture cluster's usage is: $$H(\tilde{\phi} \mid \mathcal{X}_w) = - \sum_{\tau_w \in T_w} \frac{\text{count}(\tilde{\phi}, \tau_w)}{|\tau_w|} \log_{N_{\tilde{\phi}}} \left( \frac{\text{count}(\tilde{\phi}, \tau_w)}{|\tau_w|} \right) \qquad (9)$$ This measures the diversity of the fracture cluster's usage across tasks. |
| $\alpha, \beta$ | Parameters of the Beta distribution used in Bayesian estimation; set to $\alpha = 1$, $\beta = 1$ for an uninformative prior. |
| $\zeta_n$ | Appearance indicator for task $n$; $\zeta_n = 1$ if any fracture in $\tilde{\phi}$ appears in trajectory $\tau_n$, else 0. |
| $Z$ | Set of all Fracture Cluster Options (FraCOs). |
| $z$ | A single FraCO; an option derived from a fracture cluster. |
| | *Continued on next page...* |

| Term/Symbol | Definition/Derivation |
|---|---|
| $I_z$ | Initiation set of FraCO $z$; states where $z$ can be initiated: $$I_z = \{s \in S \mid G_{z,s} \neq \emptyset\}$$ |
| $\beta_z$ | Termination condition of FraCO $z$; $z$ terminates when all actions in the selected fracture have been executed or when no matching fracture is found: $$\beta(s) = \begin{cases} 1 & \text{if all actions in } \phi_z \text{ have been executed} \\ 1 & \text{if next action is z' and } G_{z',s} = \emptyset \\ 0 & \text{otherwise} \end{cases}$$ |
| $\pi_z$ | Policy of FraCO $z$; defines the sequence of actions when the option is active (see Algorithm 1 in the paper). |
| $G_{z,s}$ | Set of fractures assigned to cluster $z$ in state $s$: $$G_{z,s} = \{\phi \in F_s \mid P(\phi \in z) > \theta\}$$ |
| $\theta$ | Threshold hyperparameter for cluster membership; determines if a fracture belongs to a cluster based on probability. |
| $N$ | Total number of experienced tasks or samples. |
| $\mathcal{T}$ | Set of all trajectories. |
| $\tau$ | An individual trajectory from the set $\mathcal{T}$. |
| $\tau_w$ | A successful trajectory; meets a predefined success criterion. |
| $T_w$ | Set of successful trajectories. |
| $\mathcal{X}_w$ | Set of tasks corresponding to successful trajectories. |
| $A$ | Action set of the environment; may include primitive actions and options. |
| $N_c$ | Number of cluster-options (options derived from fracture clusters). |
|  |  |

## A.2 DERIVATION OF THE USEFULNESS METRIC

In this appendix, we derive the usefulness ($U$) metric for fracture clusters. This metric is used to identify which fracture clusters have the greatest potential for reuse across different tasks. Usefulness is a function of the following three factors:

1. The probability that a fracture cluster appears in any given successful task:

$$P(A(\tilde{\phi}) \in \tau_w | x_w)$$

2. The probability that a fracture cluster is selected from the set of successful fracture clusters:

$$P[A(\tilde{\phi}) | \Phi_w]$$

3. The entropy of the fracture cluster's usage across all successful tasks:

$$H(A(\tilde{\phi}) \mid \mathcal{T}_w)$$

where $A(\tilde{\phi})$ represents *any* $\phi \in \tilde{\phi}$. However, for sake of notation clarity, we drop the A for this derivation.

Usefulness ($U$) is then defined as the normalized sum of these three factors:

$$U = \frac{1}{3}\left(P[\tilde{\phi} \in \tau_s | x_s] + P[\tilde{\phi} | \Phi_s] + H(\tilde{\phi} \mid \mathcal{T}_w)\right)$$

The objective is to select fracture clusters that maximize the usefulness, i.e.,

$$\arg\max_{\tilde{\phi}} U(\tilde{\phi})$$

## A.3 Deriving $P[\tilde{\phi} \in \tau_w | x_w]$ Using Bayesian Inference

We want to model the probability that a fracture cluster $\tilde{\phi}$ appears in any given successful task, $P[\tilde{\phi} \in \tau_w | x_w]$. For each successful task $x_w$ from the set of successful tasks $\mathcal{X}_w$, the presence of fracture cluster $\tilde{\phi}$ in the corresponding trajectory $\tau_w$ is represented by a binary random variable $\zeta_w$, where:

$$\zeta_w = \begin{cases} 1 & \text{if } \tilde{\phi} \in \tau_w \text{ (fracture cluster appears in the trajectory),} \\ 0 & \text{otherwise} \end{cases}$$

The variable $\zeta_w$ is modeled as a Bernoulli random variable:

$$\zeta_s \sim \text{Bernoulli}(p)$$

where $p$ is the probability that fracture cluster $\tilde{\phi}$ appears in the trajectory $\tau_s$ of task $x_w$.

Since we are uncertain about the true value of $p$, we place a Beta distribution prior on $p$:

$$p \sim \text{Beta}(\alpha, \beta)$$

where $\alpha$ and $\beta$ are hyperparameters representing our prior belief about the likelihood of $\tilde{\phi}$ appearing in a trajectory.

Given a total of $N$ tasks, the likelihood for each observation $\zeta_w$ is:

$$P(\zeta_n | p) = p^{\zeta_n}(1 - p)^{1 - \zeta_n}$$

where $\zeta_n$ is 1 if $\tilde{\phi}$ appears in trajectory $\tau_w$ for task $x_n$, and 0 otherwise.

Using Bayes' theorem, the posterior distribution of $p$ after observing data is:

$$P(p | \zeta_1, \ldots, \zeta_N) \propto P(\zeta_1, \ldots, \zeta_N | p) P(p)$$

Substituting the likelihood and the Beta prior, we get:

$$P(p | \zeta_1, \ldots, \zeta_N) \propto \prod_{n=1}^{N} p^{\zeta_n}(1 - p)^{1 - \zeta_n} \cdot p^{\alpha - 1}(1 - p)^{\beta - 1}$$

This simplifies to:

$$P(p | \zeta_1, \ldots, \zeta_N) \propto p^{\sum_{n=1}^{N} \zeta_n + \alpha - 1}(1 - p)^{N - \sum_{n=1}^{N} \zeta_n + \beta - 1}$$

Thus, the posterior distribution for $p$ follows a Beta distribution:

$$p | \zeta_1, \ldots, \zeta_N \sim \text{Beta}(\alpha_N, \beta_N)$$

where:

$$\alpha_N = \sum_{n=1}^{N} \zeta_n + \alpha, \quad \beta_N = N - \sum_{n=1}^{N} \zeta_n + \beta$$

In our experiments, we set $\alpha = 1$ and $\beta = 1$, representing an uninformative prior.

## A.4 Deriving $P[\tilde{\phi} | \Phi_w]$

The second component of the usefulness metric, $P[\tilde{\phi} | \Phi_w]$, is the probability that fracture cluster $\tilde{\phi}$ is selected from the set of successful fracture clusters. This can be computed as the relative frequency of $\tilde{\phi}$ in the set $\Phi_w$ of successful clusters:

$$P[\tilde{\phi} | \Phi_w] = \frac{\text{count}(\tilde{\phi}, \Phi_w)}{|\Phi_w|}$$

where $\text{count}(\tilde{\phi}, \Phi_w)$ is the number of times $\tilde{\phi}$ appears in the set of successful clusters, and $|\Phi_w|$ is the total number of successful clusters.

## A.5 Deriving $H(\tilde{\phi} \mid \mathcal{T}_w)$

The entropy term $H(\tilde{\phi} \mid \mathcal{T}_w)$ measures the unpredictability or diversity of the usage of fracture cluster $\tilde{\phi}$ across successful tasks. Entropy is defined as:

$$H(\tilde{\phi} \mid \mathcal{T}_w) = - \sum_{\tau_w \in \mathcal{T}_w} p[\tilde{\phi} \mid \tau_w] \cdot \log_{N_{\tilde{\phi}}}(p[\tilde{\phi} \mid \tau_w])$$

where $p[\tilde{\phi} \mid \tau_w]$ is the proportion of times that fracture cluster $\tilde{\phi}$ appears in trajectory $\tau_w$:

$$p[\tilde{\phi} \mid \tau_w] = \frac{\text{count}(\tilde{\phi}, \tau_w)}{|\tau_w|}$$

Here, $|\tau_w|$ represents the length of the trajectory $\tau_w$, and $N_{\tilde{\phi}}$ is the total number of fracture clusters considered. The choice of logarithm base, $N_{\tilde{\phi}}$, reflects the fact that we normalize entropy relative to the number of fracture clusters.

## A.6 Expected Usefulness

Having derived the empirical estimations of the three components of usefulness we can now combine these elements to calculate the expected usefulness of each fracture cluster. The expected usefulness incorporates the posterior distribution from Bayesian inference for $P[\tilde{\phi} \in \tau_w | x_s]$, as well as the empirical counts for the other components.

Thus, the expected usefulness for each fracture cluster is calculated as:

$$E[U(\tilde{\phi})] = \frac{1}{3}\left(\frac{\sum_{n=1}^{N}\zeta_n + \alpha}{N + \alpha + \beta} + \frac{\text{count}(\tilde{\phi}, \Phi_w)}{|\Phi_s|} - \sum_{\tau_w \in \mathcal{T}_w} \frac{\text{count}(\tilde{\phi}, \tau_w)}{|\tau_w|} \cdot \log_{N_{\tilde{\phi}}}\left(\frac{\text{count}(\tilde{\phi}, \tau_w)}{|\tau_w|}\right)\right)$$

where $\alpha$ and $\beta$ are the parameters of the beta distribution, which we set to $\alpha = 1$ and $\beta = 1$ in our experiments.

By calculating this expected usefulness, we can rank the fracture clusters according to their potential for reuse in future tasks. The ranking helps focus on fracture clusters that are more likely to appear in successful outcomes and contribute to the agent's performance across diverse scenarios.

## A.7 Environments

This section provides details on the environments used in our experiments, including standard grid-world domains (Four Rooms, Grid, Ramesh Maze), MetaGrid, and the Procgen suite. Each environment has been designed to evaluate different aspects of the agent's behaviour, such as navigation, exploration, and task performance.

### A.7.1 Grid-World Environments

We use three standard grid-world environments: Four Rooms, Grid, and Ramesh Maze. Figure 9 illustrates these environments.

In these grid-world environments, the action space is discrete, with four possible (primitive) actions:

$$A = \{0, 1, 2, 3\}$$

These actions correspond to moving Up, Down, Left, and Right, respectively. The agent's observation space is a 7x7 grid centered on itself, meaning it only observes a portion of the environment at any given time. This localized view allows the agent to learn how to navigate based on nearby features. This design is similar to MiniGrid (Chevalier-Boisvert et al., 2024), but in our case, the observations are ego-centric, always centered on the agent.

In tabular learning, the agent uses this 7x7 observation space to predict initiation states, though the Q-function is still based on absolute coordinates. We do not employ state-transition graphs in any of our experiments.

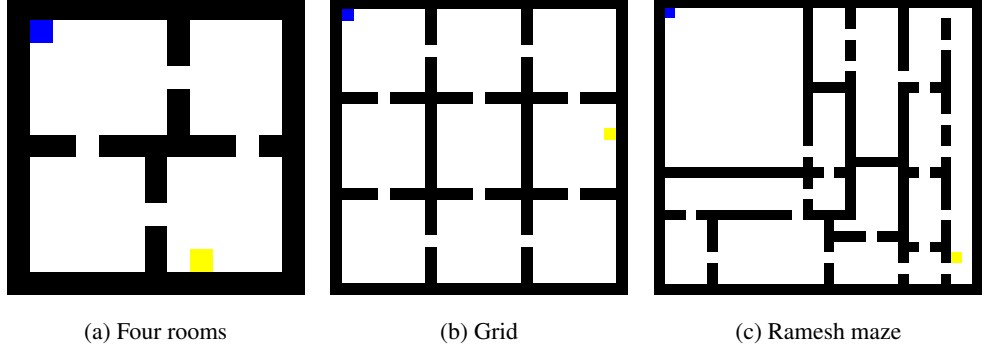

(a) Four rooms          (b) Grid          (c) Ramesh maze

Figure 9: Examples of the Four Rooms (a), Grid (Nine Rooms) (b), and Ramesh Maze (c) environments. In each, black represents walls, blue represents the agent, yellow represents the goal, and white represents empty space. The agent can move up, down, left, and right, receiving a reward upon reaching the goal. The goal's location in these figures is an example of one of many possible positions. These versions of Four Rooms, Grid and Ramesh Maze are part of the MetaGrid suite and thus have a 7x7 observation space centred on the agent.

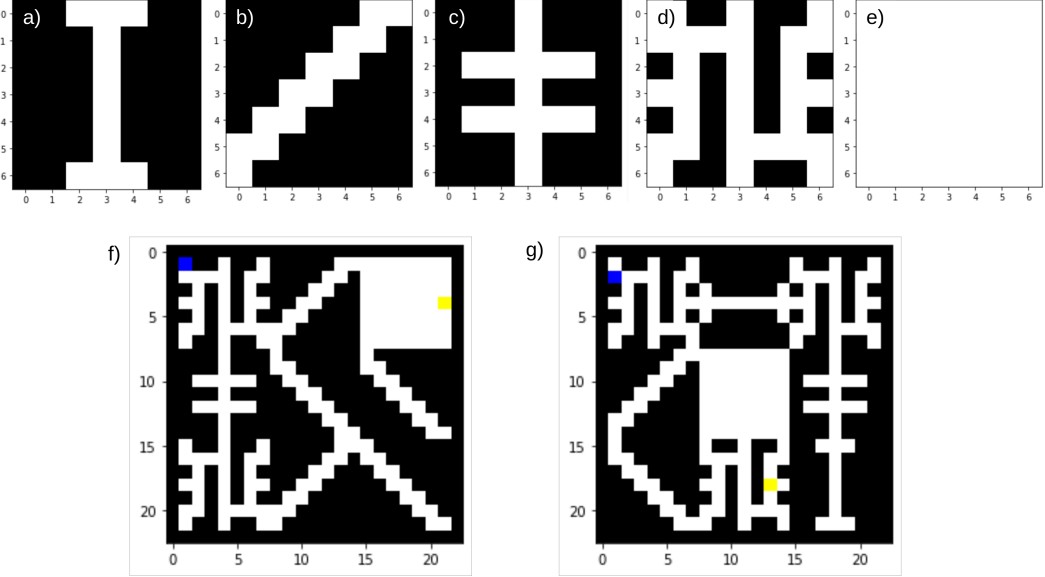

Figure 10: a) - e) demonstrate the building blocks which MetaGrid domains can be created from. f) and g) demonstrate two 21x21 configurations using these building blocks.

### A.7.2 METAGRID ENVIRONMENT

The MetaGrid environment extends the standard grid-world setup by allowing for procedurally generated maps of varying sizes. MetaGrid introduces randomness in the layout of walls and goal locations, ensuring that the agent encounters a diverse set of environments during training and evaluation. Figure 10 demonstrates the building blocks which all environments in MetaGrid are formed from, and Figure 11 shows examples of MetaGrid environments in two different sizes: 14x14 and 21x21.

The action space in MetaGrid is identical to the one used in the standard grid worlds, with four discrete actions: Up, Down, Left, and Right. Similarly, the agent observes a 7x7 grid centered on itself, allowing it to make decisions based on local information. The procedural generation in MetaGrid provides varied environments for the agent to adapt to, making it a more challenging and dynamic environment compared to static grid worlds.

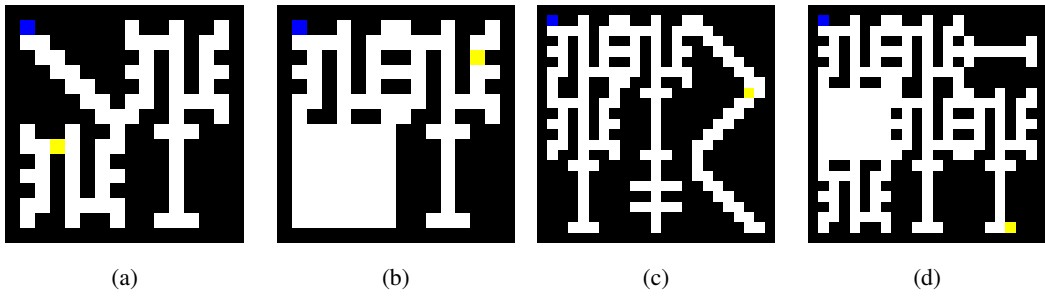

|(a)|(b)|(c)|(d)|

Figure 11: Examples of randomly generated MetaGrid environments. The blue square represents the agent, yellow represents the reward, white represents empty space, and black represents walls. Subfigures (a) and (b) show 14x14 grids, while (c) and (d) show 21x21 grids.

### A.7.3 PROCGEN ENVIRONMENTS

Procgen is a suite of procedurally generated environments designed to test generalisation and performance across diverse tasks such as navigation, exploration, combat, and puzzle-solving. Each environment provides a different variation on every reset, preventing the agent from memorizing specific layouts or solutions. Please see Cobbe et al. (2020) for the full details of these environments.

**Action Space** Procgen environments feature discrete actions like movement (up, down, left, right) and interactions (e.g., jump, shoot). Depending on the task, the action space can range from 5 to 15 actions, covering basic navigation and task-specific interactions.

**Observation Space** Unlike grid-based environments, Procgen uses 64x64 RGB pixel observations, providing rich visual input. The agent must interpret features such as walls, enemies, and obstacles to navigate and interact effectively.

**Rewards** Rewards in Procgen are sparse, given for completing tasks like reaching goals or defeating enemies. The agent must learn to explore efficiently and develop strategies for long-term success.

**Optional Parameters** To simplify learning and reduce computational cost, we activated the following Procgen parameters:

- **Distribution mode = "easy"**: Provides easier levels.

- **Use backgrounds = "False"**: Backgrounds are black to avoid additional noise.

- **Restrict themes = "True"**: Limits visual variation to a single theme, such as consistent wall styles in environments like CoinRun.

Overall, these environments—ranging from grid-world environments with discrete action spaces and ego-centric observations to the more complex Procgen environments with pixel-based observations—offer a diverse set of challenges for our agents. The combination of procedurally generated environments in MetaGrid and Procgen ensures that the agents are tested on both fixed and highly variable environments, making them suitable for evaluating the robustness of the learning algorithms used in our experiments.

### A.7.4 UMAP VISUALISATIONS OF OTHER ENVIRONMENTS

The structure observed in the latent projections of clustered Fracos as seen in Section 4.1 is also demonstrated in trained agents of other simple environments; Figure 12 visualises fracture structures in Grid (Nine rooms), CartPole, and LunarLander. The Grid environment is shown in Figure 9, CartPole and LunarLander are standard environments from the Farama Foundation Gymnasium suite Towers et al. (2023).

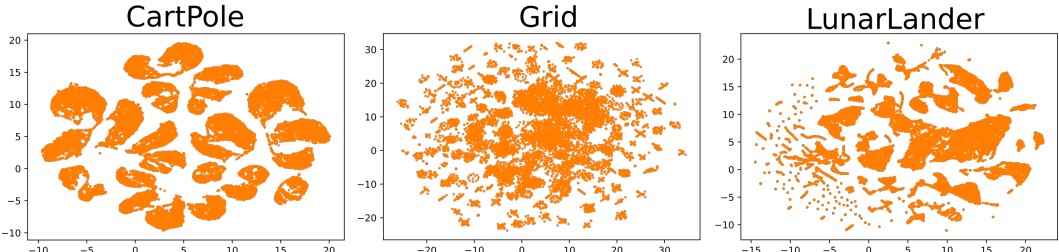

Figure 12: Two dimensional visualisations of the fractures formed for trained agents in CartPole, Grid (Nine Rooms) and LunarLander .

### A.8 FraCOs Chain Lengths and Depth Limits

In our FraCOs method, the process of matching clusters involves conducting a discrete search over potential action permutations. This process requires forming all possible permutations of actions and passing them through the saved clusterer to determine which permutation is currently viable for the meta-policy to choose. The complexity of this search is captured by the permutation formula:

$$^{B}P_{N_c} = \frac{N_c!}{(N_c - B)!}$$

where $B$ is the length of the action chain, and $N_c$ is the total number of cluster-options. As a result, the time complexity for performing this search grows factorially, $O\left(\frac{N_c!}{(N_c - B)!}\right)$, as more cluster-options are introduced or when the chain length is increased. This rapidly becomes computationally expensive as these numbers increase.

#### A.8.1 Chain Length and Depth in Tabular Experiments

Due to this factorial growth, it is crucial to limit the number of cluster-options and the chain length in experiments that involve discrete cluster search (Experiments 1, 2 and 3.1). For all experiments using cluster search, we chose a **chain length of 2** to keep the computational complexity manageable. Additionally, we restricted the depth of the FraCOs hierarchy to **3 or 4 levels**, depending on the complexity of the environment.

This limitation on depth and chain length helps maintain a balance between the richness of the learned options and the feasibility of performing the cluster search in a reasonable amount of time. The factorial growth of the search process becomes prohibitive as more cluster-options or deeper chains are introduced, making this constraint necessary for efficient execution of our experiments.

#### A.8.2 Why This is not a Problem for Neural Network Cluster Predictions

In contrast, when we extend the FraCOs initiation and action estimation to be used with neural networks, the limitation of conducting computationally expensive permutation searches is alleviated. Neural networks can learn initiation sets and make predictions in a continuous manner, bypassing the need for discrete cluster searches. This removes the necessity of factorially growing search complexity, allowing for more flexibility in chain lengths and depth.

However, the challenge in neural network experiments lies in the sheer number of timesteps required for training and evaluation. For example, in our deep experiments (at a depth of two), training involved many millions of timesteps across nine different environments, totaling **180 million steps** for pre-training, repeated for three seeds. Testing required an additional **120 million steps**, also repeated for three seeds. This was only for FraCOs. When we include experiments using OC (Option Critic), PPO, and PPO25, the total number of timesteps across all experiments becomes **3.06 billion**.

To manage these computational demands, we used a **chain length of 3**. This allowed us to conduct only two warm-up phases, while ensuring that options could still be executed for a reasonable

duration (a maximum of nine steps, i.e., 3x3). This setup enabled us to complete the necessary pre-training and testing without sacrificing the quality of the learned options while maintaining computational feasibility.

## A.9 FRACOS EXPERIMENT PARAMETERS FOR TABULAR METHODS

Tabular Q-Learning is a reinforcement learning algorithm where the agent learns the optimal action-value function $Q(s, a)$, which estimates the expected cumulative reward for taking action $a$ in state $s$ and following the optimal policy thereafter. The agent interacts with the environment, updates the Q-values for each state-action pair based on rewards, and converges to the optimal policy over time Sutton & Barto (2018). We implement a vectorized Q learning method.

The key hyperparameters used in all Tabular Q-Learning experiments are listed in Table 2.

Table 2: Hyperparameters for Tabular Q-Learning Experiment

| Hyperparameter | Value | Description |
|---|---|---|
| eps | 0.1 | Exploration rate for $\epsilon$-greedy policy. Determines the probability of taking a random action instead of the action with the highest Q-value. |
| alpha | 0.1 | Learning rate for Q-value updates. Controls how much the Q-value is updated in each iteration. |
| gamma | 0.99 | Discount factor for future rewards. |
| num_steps | 64 | Number of steps per episode before environment reset. |
| max_ep_length | 1000 | Maximum timesteps allowed in an episode. |
| anneal_lr | True | Whether to anneal the learning rate as training progresses. |
| batch_size | 64 | Number of state-action-reward tuples processed in a batch. |
| Number of Envs | 64 | Number of vectorized environments |

In Tabular Q-Learning, the agent repeatedly updates its Q-values for each state-action pair, gradually converging to the optimal policy. By balancing exploration and exploitation, adjusting the learning rate, and prioritizing long-term rewards, the agent learns to optimize its decision-making in the given environment.

**FraCOs (Fracture Cluster Options) for Tabular Methods:** For reproducibility, the key hyperparameters used in the clustering process and other implementation details are outlined below.

In all tabular experiments, we use HDBSCAN as the clustering method (Campello et al., 2013). The clustering hyperparameters are:

Table 3: Clustering Hyperparameters for FraCOs

| Hyperparameter | Value |
|---|---|
| Chain length (b) | 2 |
| Minimum cluster size | 15 |
| Metric | Euclidean |
| Minimum samples | 1 |
| Generate minimum spanning tree | True |

The following table lists the minimum success rewards required for each environment:

**FraCOs-Specific Hyperparameters:**

The generalisation strength represents the threshold a fracture cluster must pass to be considered for initiation. The threshold is defined as $1 -$ HDBSCAN.predict.strength $>$ generalisation strength.

The FraCOs bias factor determines how much initial Q-values should be scaled to encourage transfer, similar to optimistic initial values. For FraCOs bias depth annealing, the bias depth increases with

Table 4: Minimum Success Reward per Environment

| Environment | Minimum Success Reward |
|---|---|
| Four Rooms | 0.97 |
| Grid | 0.60 |
| Ramesh Maze | 0.70 |
| MetaGrid 14x14 | 0.95 |

Table 5: FraCOs Hyperparameters

| Hyperparameter | Value |
|---|---|
| Generalisation strength | 0.01 |
| FraCOs bias factor | 100 |
| FraCOs bias depth anneal | True |

deeper FraCOs. For example, with a bias factor of 100 and depth of 3, the first bias is scaled by the cube root of 100, the second by the square root, and the final by 100.

### A.10 FraCOs Modifications for Deep Learning

In our experiments with deep learning, particularly in environments with large state spaces such as the 64x64x3-dimensional Procgen environments, we found that **HDBSCAN** failed to accurately capture meaningful clusters or predict clusters effectively. Initially, we attempted to integrate a **Variational Autoencoder (VAE)** to use its latent space representation in the fracture formation process Kingma & Welling (2013). However, clustering methods still struggled to deliver satisfactory results.

Consequently, we adopted a simpler clustering approach and shifted to using neural networks to predict both the initiation states and the policy.

**Simpler Clustering**. To simplify the clustering process, we based clusters solely on sequences of actions. For instance, with a chain length of three, any fracture formed by the action sequence "up", "up", "right" was clustered together, independent of the state. While this approach overlooks some intricacies captured by state-based fracture formations, it was necessary to handle the increased complexity of environments like Procgen.

**Cluster Selection**. The cluster selection process remained unchanged. We continued to use the usefulness metric, as defined in Equation 3, to select clusters.

**Initiation Prediction**. Instead of relying on clustering methods to predict initiation states, we trained a neural network to predict the states corresponding to each fracture cluster. This process involved two steps:

1. First, we trained a **Generative Adversarial Network (GAN)** to augment the states in each fracture cluster (since neural networks typically require large datasets).

2. Using both the real and generated states, we trained a neural network as a classifier to predict which fracture cluster a given state belonged to. One neural network was trained to predict all initiations at each hierarchical level. This method significantly improved efficiency, reducing the need for permutation-based discrete searches to a single forward pass.

**Policy Prediction**. For policy prediction, we utilized a shortcut. Since all FraCOs are derived from trajectories generated by a pre-trained agent, the policy of this trained agent already serves as an approximation for the FraCO policy. We saved the agent's policy at the end of trajectory generation and used it as the policy for all FraCOs. This reuse of the agent's policy minimized additional computation without sacrificing accuracy.

**Termination Condition**. The termination condition for each FraCO was determined solely by the chain length, maintaining a fixed execution limit per option.

A.11    EXPERIMENT PARAMETERS FOR DEEP METHODS

All deep methods in our experiments were based on CleanRL's implementation of PPO for the Procgen environments, and we used the same hyperparameters from this implementation. This ensured that the PPO baseline was hyperparameter-tuned, providing a strong and well-optimized baseline for comparison. However, FraCOs and OC-PPO were not specifically hyperparameter-tuned for these environments. Despite this, we consider it reasonable to assume that FraCOs and OC-PPO would perform near their best, as they share similar PPO update mechanisms and underlying structures with the baseline PPO.

While there exists an implementation of OC-PPO by Klissarov et al. (2017), we opted to implement our own version to maintain consistency with CleanRL's PPO implementation. This approach was necessary to ensure that any observed differences in performance were due to algorithmic design rather than implementation differences.

We chose OC-PPO over the standard Option Critic for two primary reasons. First, in our experiments with the standard Option Critic, we observed that the options collapsed quickly, converging to the same behaviour. By integrating the entropy bonus provided by the PPO update, we were able to alleviate this collapse and maintain more diverse option behaviours. Second, PPO has demonstrated significantly better performance than Advantage Actor Critic (A2C) (Mnih, 2016) in the Procgen environments. Given that the original Option Critic framework is based on A2C, using it as a baseline would have led to an unfair comparison, as A2C has been shown to be less effective in these environments. Therefore, incorporating PPO in both FraCOs and OC-PPO allowed for a fairer and more balanced comparison.

We outline the hyperparameters which we used in the implementation below. All code will be provided from the authors github upon publication.

A.11.1    PPO

We use CleanRL's Procgen implementation as our baseline Huang et al. (2022). The only adaption we make is that we implement entropy annealing, we also have some wrappers which mean Procgen can be used with the Gymnasium API. We have another wrapper which is used for handling multi-level FraCOs in vectorized environments. These wrappers are also applied to the baseline PPO.

**Hyperparameters** Table 6 provide hyperparameters for PPO.

| Parameter | Value | Explanation |
|---|---|---|
| easy | 1 | 1 activates "easy" Procgen setting |
| gamma | 0.999 | The discount factor. |
| vf_coef | 0.5 | Coefficient for the value function loss. |
| ent_coef | 0.01 | Entropy coefficient. |
| norm_adv | true | Whether to normalize advantages. |
| num_envs | 64 | Number of parallel environments. |
| anneal_lr | true | Whether to linearly anneal the learning rate. |
| clip_coef | 0.1 | Clipping coefficient for the policy objective in PPO. |
| num_steps | 256 | Number of *decisions* per environment per update. |
| anneal_ent | false | Whether to anneal the entropy coefficient over time. |
| clip_vloss | true | Whether to clip the value loss in PPO. |
| gae_lambda | 0.95 | The lambda parameter for Generalized Advantage Estimation (GAE). |
| proc_start | 1 | Indicates the starting level for Procgen environments. |
| learning_rate | 0.0005 | The learning rate for the PPO optimizer. |
| max_grad_norm | 0.5 | Maximum norm for gradient clipping. |
| update_epochs | 2 | Number of epochs per update. |
| num_minibatches | 8 | Number of minibatches. |
| max_clusters_per_clusterer | 25 | The maximum number FraCOs per level. |

Table 6: Selected parameters for the FraCOs implementation with PPO

### A.11.2 FRACOS

**Neural Network architectures.**

- **Input:** $c \times h \times w$ (image observation).
- **Convolutional Layer 1:** 16 filters, kernel size $3 \times 3$, stride 1, padding 1, followed by ReLU activation.
- **Convolutional Layer 2:** 32 filters, kernel size $3 \times 3$, stride 1, padding 1, followed by ReLU activation.
- **Convolutional Layer 3:** 32 filters, kernel size $3 \times 3$, stride 1, padding 1, followed by ReLU activation.
- **Flatten Layer:** Converts the output of the last convolutional layer into a 1D tensor.
- **Fully Connected Layer:** 256 units, followed by ReLU activation.
- **Actor Head:** A linear layer with 256 input units and `total_action_dims` output units, initialized with a standard deviation of 0.01.
- **Critic Head:** A linear layer with 256 input units and 1 output unit (for the value function), initialized with a standard deviation of 1.

The model consists of two heads:

- **Actor Head:** Outputs a probability distribution over the action space for the agent to select actions.
- **Critic Head:** Outputs the value function, which estimates the expected return for the current state.

The network uses ReLU activations after each convolutional and fully connected layer, and the actor and critic heads share the same convolutional layers but have distinct fully connected output layers. In the FraCOs-SSR implementation, the convolutional layers are not reset after the warm-up phase.

**Shared State Representation (SSR) details.**

In the above architecture, both the actor and critic heads share a common set of convolutional layers. These shared layers process the raw image observations from the environment and extract useful spatial features that are fed into both the actor and critic branches. The use of shared convolutional layers allows the model to leverage the same learned feature representations for both policy and value estimation, promoting efficiency and consistency in learning.

In the **FraCOs-SSR** implementation, the shared convolutional layers are trained during the initial warm-up phase, but they are not reset afterward. This allows the network to retain its learned feature representations across multiple tasks and reuse them for both policy and value estimation during subsequent training phases. By freezing these convolutional layers after the warm-up phase, the network preserves its ability to generalize, while the distinct fully connected layers in the actor and critic heads continue to adapt to new tasks.

**Hyperparameters**

Table 7 provides the full list of FraCOs hyperparameters in experimentation.

| Parameter | Value | Explanation |
|---|---|---|
| easy | 1 | 1 activates "easy" Procgen setting |
| gamma | 0.999 | The discount factor. |
| vf_coef | 0.5 | Coefficient for the value function loss. |
| ent_coef | 0.01 | Entropy coefficient. |
| norm_adv | true | Whether to normalize advantages. |
| num_envs | 64 | Number of parallel environments. |
| anneal_lr | true | Whether to linearly anneal the learning rate. |
| clip_coef | 0.1 | Clipping coefficient for the policy objective in PPO. |
| num_steps | 128 | Number of *decisions* per environment per update. |
| anneal_ent | true | Whether to anneal the entropy coefficient over time. |
| clip_vloss | true | Whether to clip the value loss in PPO |
| gae_lambda | 0.95 | The lambda parameter for Generalized Advantage Estimation (GAE). |
| proc_start | 1 | Indicates the starting level for Procgen environments. |
| learning_rate | 0.0005 | The learning rate for the PPO optimizer. |
| max_grad_norm | 0.5 | Maximum norm for gradient clipping |
| update_epochs | 2 | Number of epochs per update. |
| num_minibatches | 8 | Number of minibatches |
| max_clusters_per_clusterer | 25 | The maximum number FraCOs per level |

Table 7: Selected parameters for the FraCOs implementation with PPO

### A.11.3 OC-PPO

**Architectures**

- **Input**: $c \times h \times w$ (image observation).
- **Convolutional Layer 1**: 32 filters, kernel size 3×3, stride 2, followed by ReLU activation.
- **Convolutional Layer 2**: 64 filters, kernel size 3×3, stride 2, followed by ReLU activation.
- **Convolutional Layer 3**: 64 filters, kernel size 3×3, stride 2, followed by ReLU activation.
- **Flatten Layer**: Converts the output of the last convolutional layer into a 1D tensor.
- **Fully Connected Layer**: 512 units, followed by ReLU activation.

The model consists of several heads:

- **Option Selection Head**: A linear layer with 512 input units and $num\_options$ output units (for selecting options), initialized with orthogonal weight initialization and a bias of 0.0.
- **Intra-Option Action Head**: A linear layer with 512 input units and $num\_actions$ output units (for selecting actions within an option), initialized with orthogonal weight initialization and a bias of 0.0.
- **Critic Head**: A linear layer with 512 input units and 1 output unit (for the value function), initialized with orthogonal weight initialization and a standard deviation of 1.
- **Termination Head**: A linear layer with 512 input units and 1 output unit (for predicting termination probabilities), followed by a sigmoid activation to output a probability between 0 and 1.

The architecture is designed to share a common state representation across different heads (option selection, intra-option action selection, value estimation, and option termination). Each head uses the shared state representation for their specific outputs:

- **Option Selection Head**: Outputs a probability distribution over available options.
- **Intra-Option Action Head**: Outputs a probability distribution over the primitive actions available within the current option.

- **Critic Head**: Outputs the value function, estimating the expected return for the current state.
- **Termination Head**: Outputs a termination probability for each option, determining whether the agent should terminate the option at the current state.

**Hyperparameters**. Table 8 provide the hyperparameters used in OC-PPO.

| Parameter | Value | Explanation |
|---|---|---|
| easy | 1 | Activates the "easy" Procgen setting. |
| gamma | 0.999 | The discount factor for future rewards. |
| vf_coef | 0.5 | Coefficient for the value function loss in PPO. |
| norm_adv | true | Whether to normalize advantages before policy update. |
| num_envs | 32 | Number of parallel environments for training. |
| anneal_lr | true | Linearly anneals the learning rate throughout training. |
| clip_coef | 0.1 | Clipping coefficient for the PPO policy objective. |
| num_steps | 256 | Number of steps per environment before an update is performed. |
| anneal_ent | true | Whether to anneal the entropy coefficient over time. |
| clip_vloss | false | Whether to clip the value loss in PPO updates. |
| gae_lambda | 0.95 | Lambda for Generalized Advantage Estimation (GAE). |
| proc_start | 1 | Indicates the starting level for the Procgen environment. |
| num_options | 25 | Number of options learned by the agent. |
| ent_coef_action | 0.01 | Coefficient for the entropy of the action policy. |
| ent_coef_option | 0.01 | Coefficient for the entropy of the option policy. |
| learning_rate | 0.0005 | Learning rate for the PPO optimizer. |
| max_grad_norm | 0.1 | Maximum norm for gradient clipping. |
| update_epochs | 2 | Number of epochs per PPO update. |
| num_minibatches | 4 | Number of minibatches per PPO update. |

Table 8: Selected hyperparameters for OC-PPO implementation

### A.12 OC-PPO UPDATE MECHANISM

The Option-Critic with PPO (OC-PPO) extends the standard Proximal Policy Optimization (PPO) algorithm by incorporating hierarchical options through the Option-Critic (OC) framework. The following key components distinguish OC-PPO from the standalone PPO and OC implementations:

**1. Separate Action and Option Policy Updates:** In OC-PPO, two sets of policy updates are performed: one for the action policy within an option and one for the option selection policy. Both policies are optimized using the clipped PPO objective, but they operate at different levels of the hierarchy:

- *Action Policy:* The action policy selects the primitive actions based on the current option. For each option, the log-probabilities of actions are calculated, and the advantage function is used to update the action policy.
- *Option Policy:* The option policy determines which option should be selected at each state. This option selection is also updated using the PPO objective, with its own log-probabilities and advantage terms.

Both the action and option policies are clipped to prevent overly large updates, following the standard PPO procedure:

$$\text{Loss}_{\text{policy}} = \max\left( A(\pi) \cdot \frac{\pi_{\text{new}}}{\pi_{\text{old}}}, A(\pi) \cdot \text{clip}\left( \frac{\pi_{\text{new}}}{\pi_{\text{old}}}, 1 - \epsilon, 1 + \epsilon \right) \right)$$

Here, $\pi_{\text{new}}$ and $\pi_{\text{old}}$ represent the new and old policies for both actions and options, and $A(\pi)$ is the advantage function. This clipping is applied separately for both action and option updates, providing stability in training.

**2. Shared State Representation for Action and Option Policies:** The OC-PPO architecture shares the state representation between the action and option policies but maintains distinct linear layers for each policy. The state representation is learned via a shared convolutional network. This shared representation ensures that both action and option policies are informed by the same state encoding, allowing for consistent hierarchical decision-making.

- *Action Policy Head:* Receives the state representation and outputs the action logits for the current option.
- *Option Policy Head:* Receives the state representation and outputs the option logits for option selection.

**3. Termination Loss for Options:** A unique component of OC-PPO is the termination loss, which encourages the agent to decide when to terminate an option and select a new one. The termination function outputs the probability that the current option should terminate. This probability is combined with the advantage function to compute the termination loss:

$$\text{Loss}_{\text{termination}} = \mathbb{E}[\text{termination\_probability} \cdot (\text{return} - \text{value})]$$

The termination loss is minimized when the agent terminates the option appropriately, i.e., when the return associated with continuing the current option is less than the estimated value of switching to a new option. The termination probability is computed by a separate network head from the shared state representation.

**4. Hierarchical Advantage Calculation:** OC-PPO calculates separate advantage terms for actions and options:

- *Action Advantage:* Based on the immediate rewards from the environment while following the current option's policy.
- *Option Advantage:* Based on the value of switching to a new option versus continuing with the current option.

Each advantage is normalized independently, and separate PPO updates are applied to both the action and option policies based on their respective advantage functions.

**5. Regularization via Entropy for Both Action and Option Policies:** As in standard PPO, entropy regularization is applied to encourage exploration. However, in OC-PPO, this regularization is applied both at the action level (to encourage diverse action selection within an option) and at the option level (to encourage exploration of different options). The overall loss function includes separate entropy terms for actions and options:

$$\text{Loss}_{\text{entropy}} = \alpha_{\text{action}} \cdot \mathbb{H}(\pi_{\text{action}}) + \alpha_{\text{option}} \cdot \mathbb{H}(\pi_{\text{option}})$$

**6. Clipping for Value Function:** Like in PPO, OC-PPO also employs clipping for the value function updates to prevent large changes in the value estimate between consecutive updates. This applies to the shared value function, which evaluates the expected returns from both primitive actions and options.

$$\text{Loss}_{\text{value}} = 0.5 \cdot \max\left((V_{\text{new}} - R)^2, (V_{\text{old}} + \text{clip}(V_{\text{new}} - V_{\text{old}}, -\epsilon, \epsilon))^2\right)$$

## A.13 HYPERPARAMETER SWEEPS

In this appendix we provide evidence of hyperparameter sweeps for both OC-PPO and HOC. We use the hyperparameters found for OC-PPO in our final results for Procgen. We decided not to use HOC after having difficulty with option collapse.

We conduct all hyperparameter sweeps for 2 million time-steps on a the Procgen environments of Fruitbot, Starpilot and Bigfish. OC-PPO had 105 total experiments and HOC had 170.

### A.13.1 OC-PPO

In Figure 13 we visualise all experiments conducted over all seeds for learning rates [1e-3, 1e-4, 1e-5] and entropy coefficients of [1e-1, 1e-2, 1e-3]. In Table 9 we state the averaged results. We decided on 0.01 for the learning rate and 0.001 for the entropy coefficient.

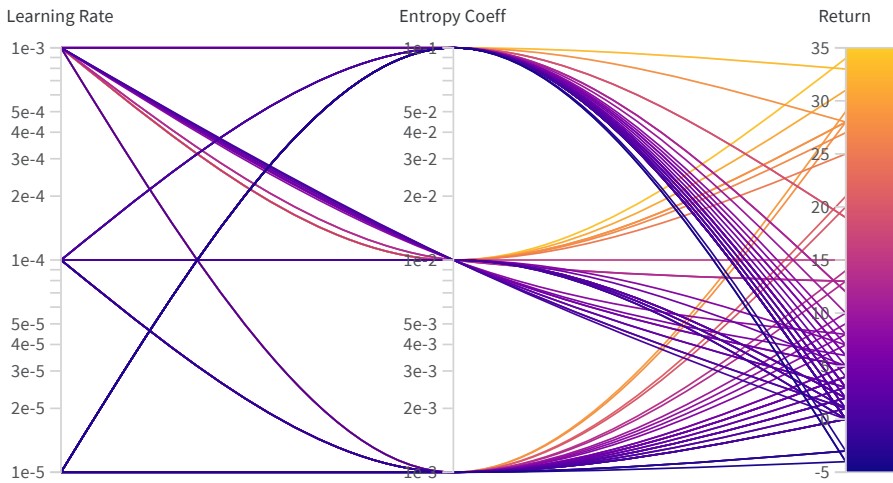

Figure 13: Parallel coordinates plot demonstrating effects of hyperparameters. This is for OC-PPO

| Learning Rate | Entropy Coefficient | Final Returns |
|---|---|---|
| 0.00001 | 0.83 | |
| | 0.001 | 0.79 |
| | 0.1 | 0.86 |
| 0.0001 | 3.2 | |
| | 0.001 | 5.67 |
| | 0.01 | 1.92 |
| | 0.1 | 0.83 |
| 0.001 | 12.67 | |
| | 0.001 | 15.83 |
| | 0.01 | 12.22 |
| | 0.1 | 11.75 |

Table 9: Results for different learning rates and entropy coefficients for OC-PPO.

## A.14 FULL PROCGEN RESULTS

Learning curves across all tested Procgen environments and shown in Figure 14

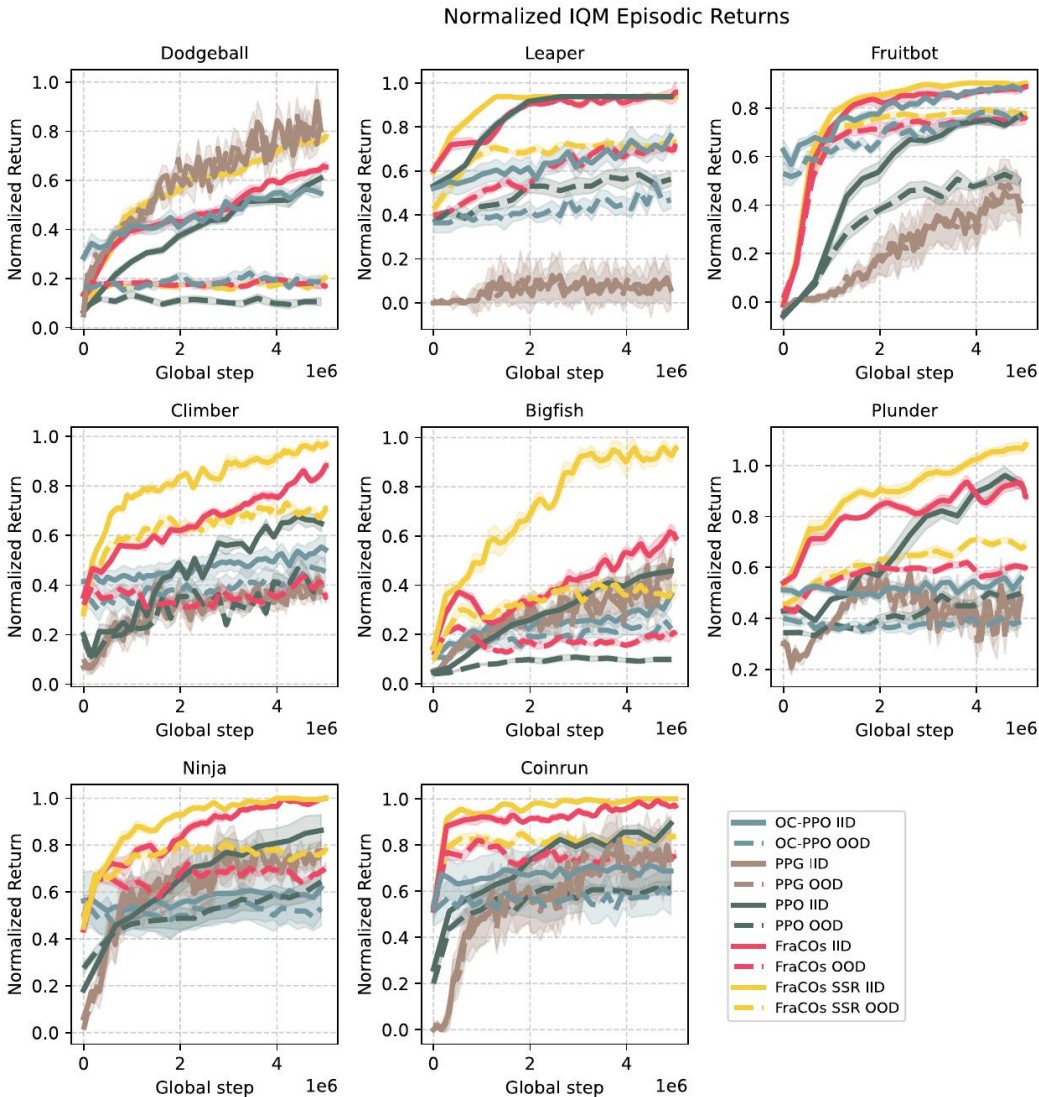

Figure 14: Learning curves for all methods on all Procgen environments

## A.15 SUCCESS CRITERIA HYPERPARAMETERS

Table 10 states all success criteria used in our work.

| Environment | Minimum Success Returns |
|---|---|
| Four Rooms | 0.97 |
| Nine Rooms | 0.60 |
| Ramesh Maze | 0.70 |
| MetaGrid 14x14 | 0.95 |
| BigFish | 5 |
| Climber | 7 |
| CoinRun | 7.5 |
| Dodgeball | 5 |
| FruitBot | 7.5 |
| Leaper | 7 |
| Ninja | 7.5 |
| Plunder | 10 |

Table 10: Minimum Success Returns for Various Environments

### A.16 CLUSTERING ANALYSIS

We compared different clustering methods and hyperparameters to ensure we were using a sensible combination. We desired our clustering method to have prediction functionality for new data and we didn't want to specify the number of clusters. The only sensible clustering method which remained was HDBSCAN. Regardless we analysed others to ensure HDBSCAN was not significantly worse.

We first generated trajectories in the Nine Rooms environment, created embeddings using UMAP and then conducted clustering with various methods and parameters. Figures 15 – 20 provide visualisations of the results. We decided on HDBSCAN with minimum size of 15 and Eucledian distance metrics.

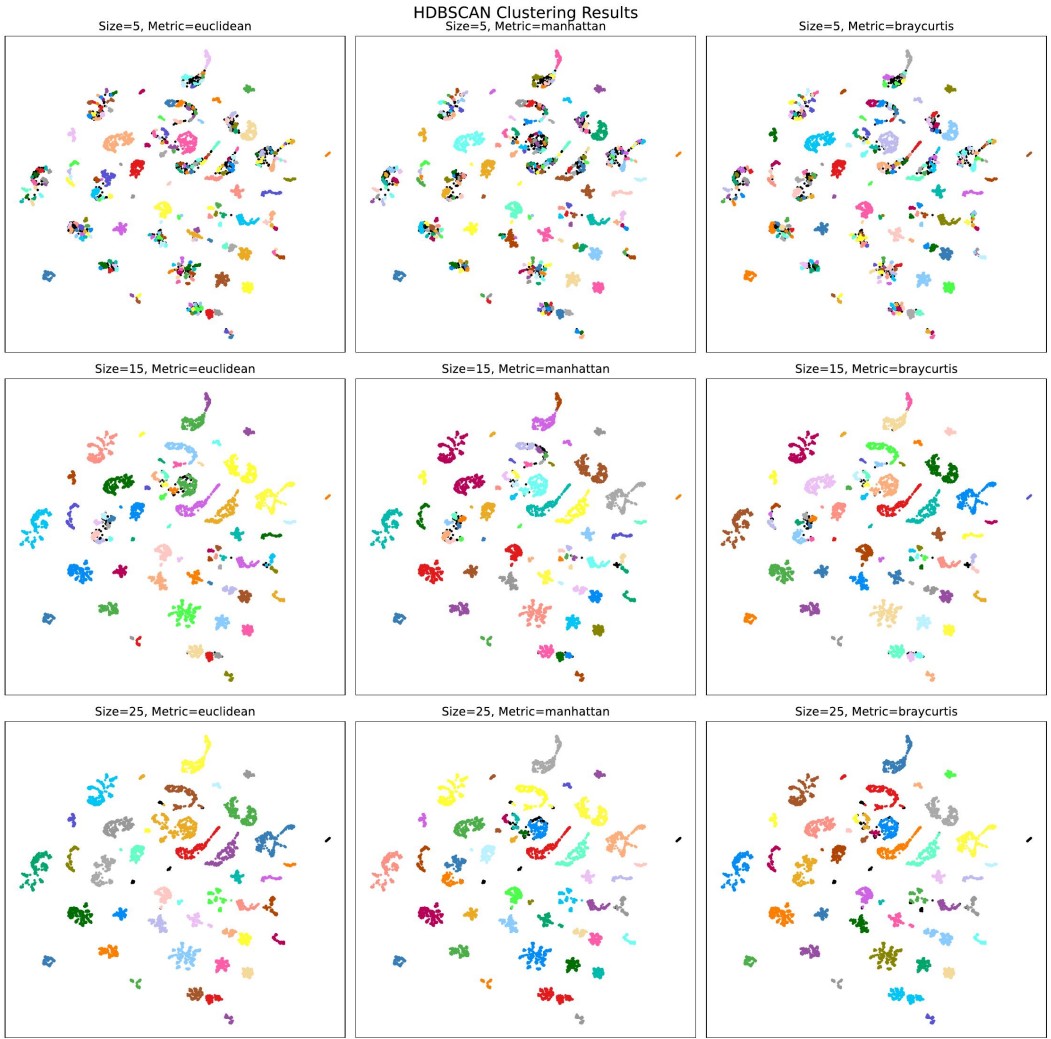

Figure 15: HDBSCAN clustering comparison

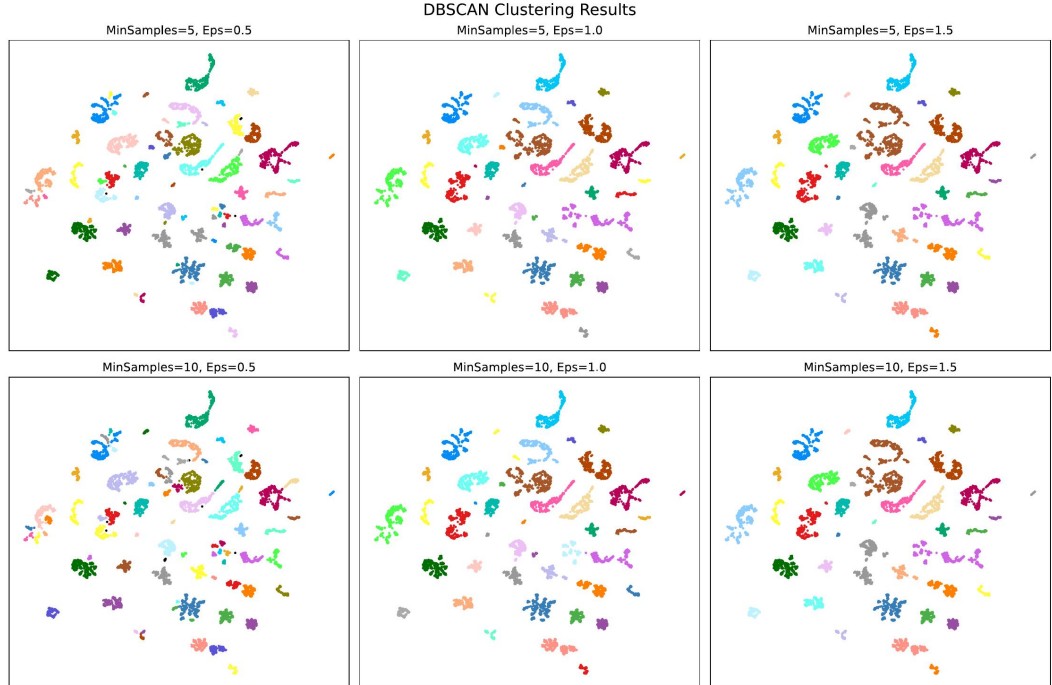

Figure 16: DBSCAN clustering comparison

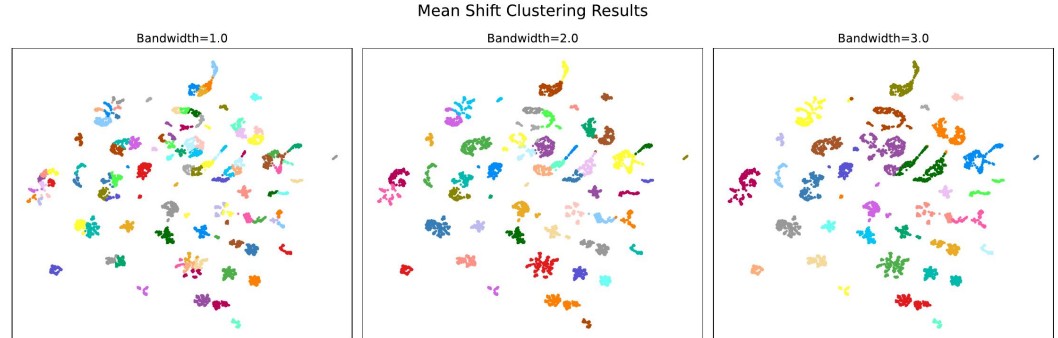

Figure 17: Mean Shift clustering comparison

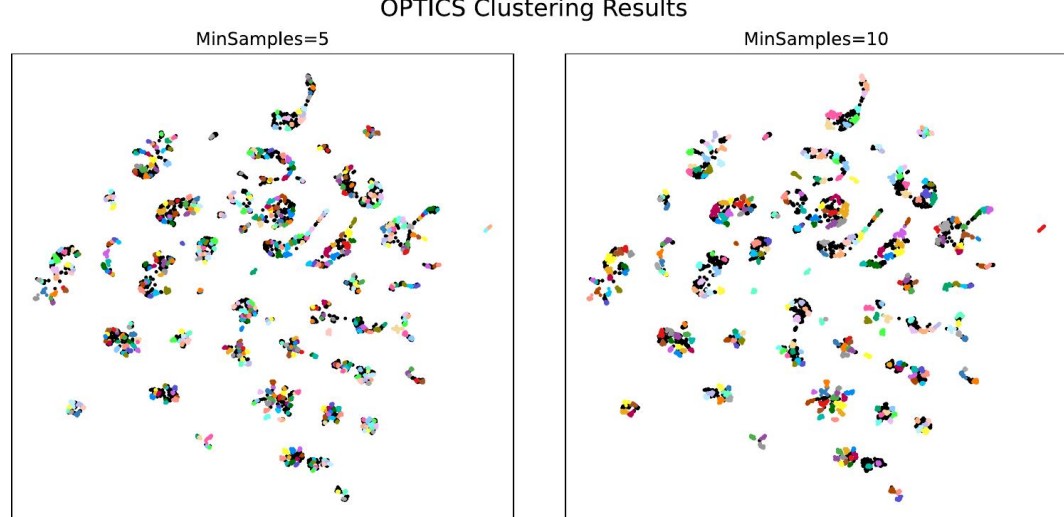

Figure 18: Optics clustering comparison

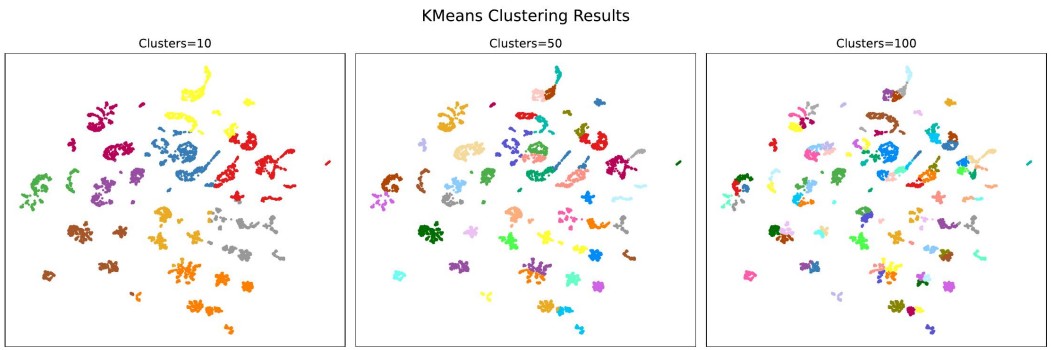

Figure 19: Kmeans clustering comparison

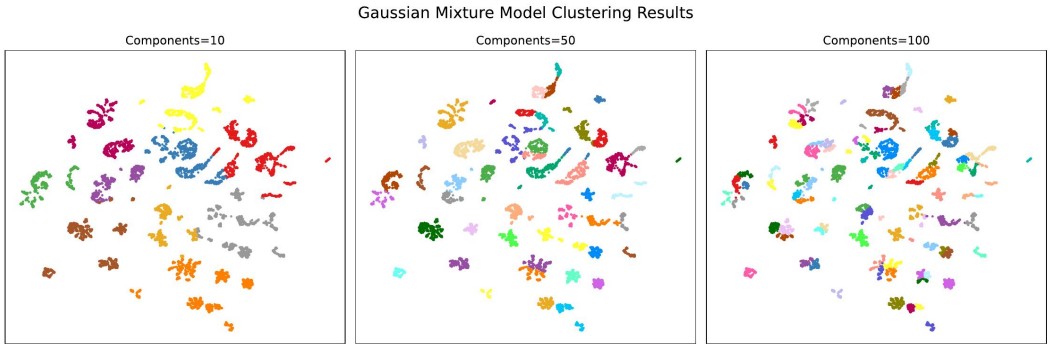

Figure 20: GMM clustering comparison

## A.17 Usefulness Weighting Qualitative Analysis

The usefulness equation assumes equal weighting of appearance probability, relative frequency, and estimated entropy. Here, we qualitatively demonstrate the effects of altering these weights. We rewrite the equation below and, in Tables 11 and 12, conduct ablation studies and visualize the top eight selected Fracture Clusters in the nine rooms environment. For this experiment, we set the success threshold to 0.97 and use a chain length of four.

The tables demonstrate that each metric selects reasonable fracture clusters. Comparing the top and bottom eight fracture clusters of each metric, we intuitively observe that the top-ranked clusters are more likely to be useful for future tasks. However, the rankings exhibit distinct differences depending on the ablations. For instance, when $A = 0$, $B = 1$, $C = 0$, the ranking prioritizes the relative frequency of selecting a fracture cluster, regardless of whether the corresponding trajectory was deemed successful. This is evident in the first-ranked fracture cluster for this configuration. With $B = 1$, the top-ranked cluster represents trajectories that frequently move away from the initial state—a common occurrence. Conversely, with $A = 1$, the selected cluster traverses a bottleneck, reflecting the prioritization of appearance in the successful trajectory. This outcome aligns with intuition since most failed trajectories also originate from the initial state, causing such clusters to rank lower when $A = 1$.

The effects of the entropy term ($C$) are harder to interpret intuitively without a detailed understanding of the full training set.

While tuning $A$, $B$, and $C$ in Equation 10 could offer benefits during the research phase, we adopted the equal-weight assumption. This approach simplifies the model's hyper-parameters and provides a baseline for future work. Adopting equal weights is not uncommon; for instance, Eysenbach et al. (2019) use equal weighting in their skill discovery objective.

$$E[U_{(\tilde{\phi})}] = \frac{1}{3} \left( A \underbrace{\frac{\sum_{n=1}^{N} \zeta_n + \alpha}{N + \alpha + \beta}}_{\text{(1) Appearance Probability}} + B \underbrace{\frac{\text{count}(\tilde{\phi}, \Phi_w)}{|\Phi_w|}}_{\text{(2) Relative Frequency}} - C \underbrace{\sum_{\tau_w \in T_w} \frac{\text{count}(\tilde{\phi}, \tau_w)}{|\tau_w|} \log_{N_{\tilde{\phi}}} \left( \frac{\text{count}(\tilde{\phi}, \tau_w)}{|\tau_w|} \right)}_{\text{(3) Estimated Entropy}} \right).$$

Table 11: Ranking of fracture clusters based on different weight configurations. Each cell contains an image visualisation of that fracture cluster.

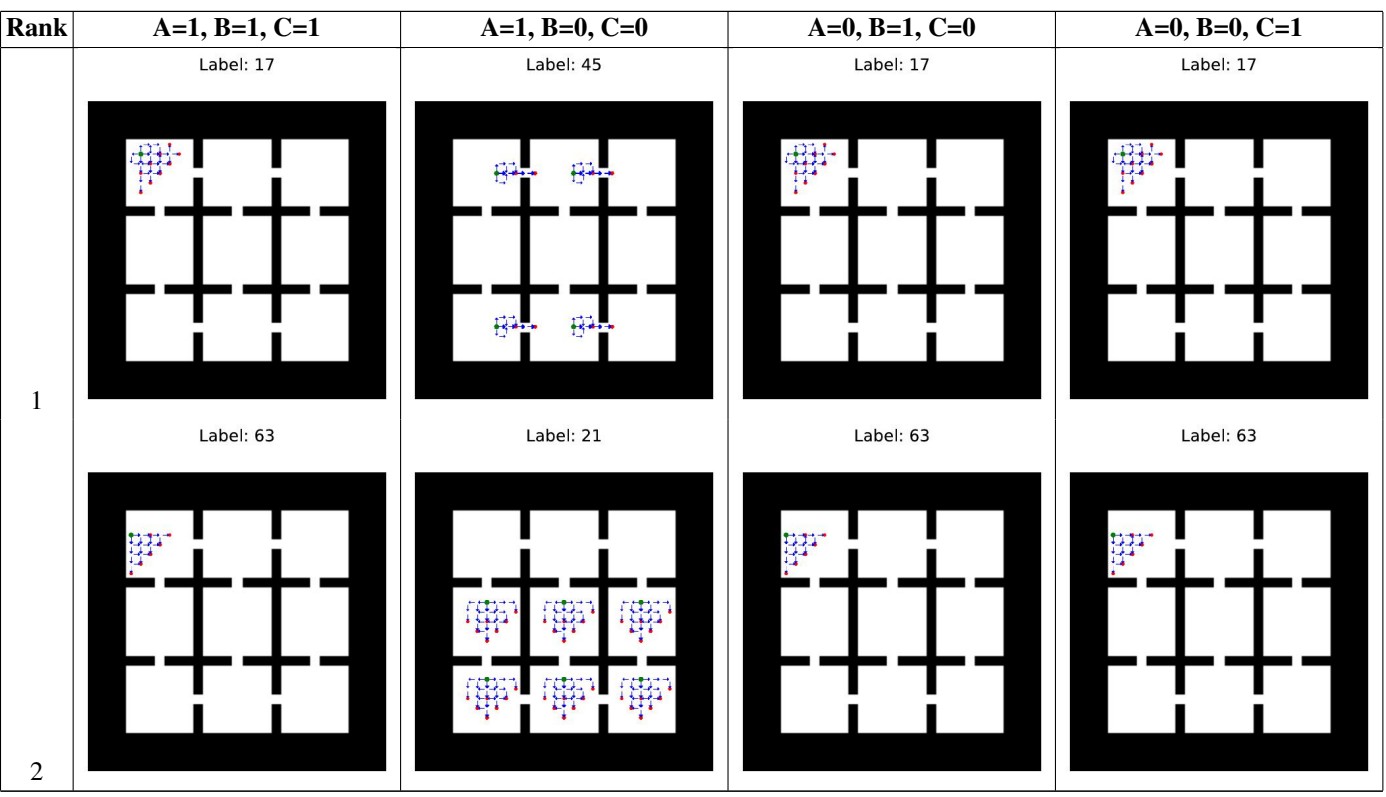

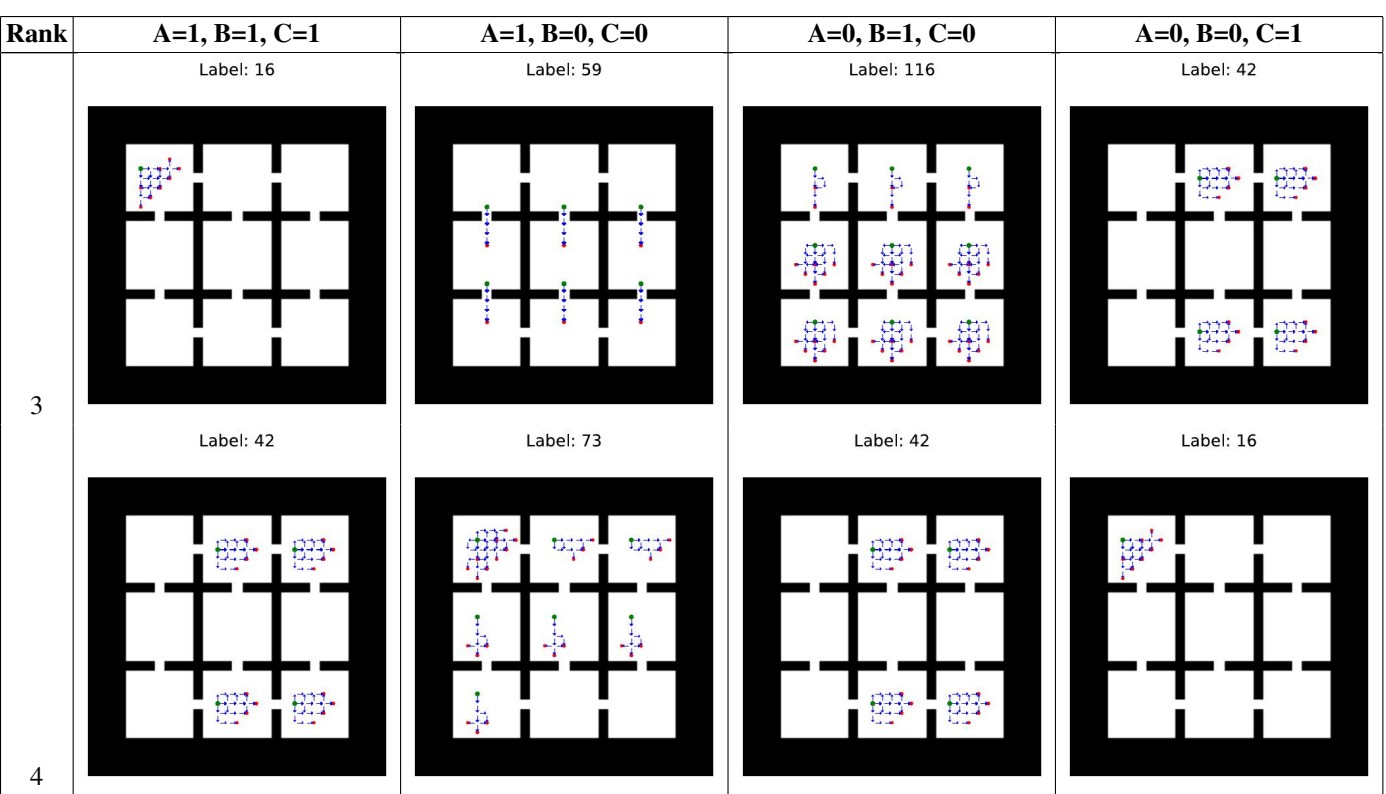

| Rank | A=1, B=1, C=1 | A=1, B=0, C=0 | A=0, B=1, C=0 | A=0, B=0, C=1 |
|------|---------------|---------------|---------------|---------------|
| 5 | Label: 116 | Label: 15 | Label: 16 | Label: 133 |
| 6 | Label: 29 | Label: 57 | Label: 29 | Label: 116 |

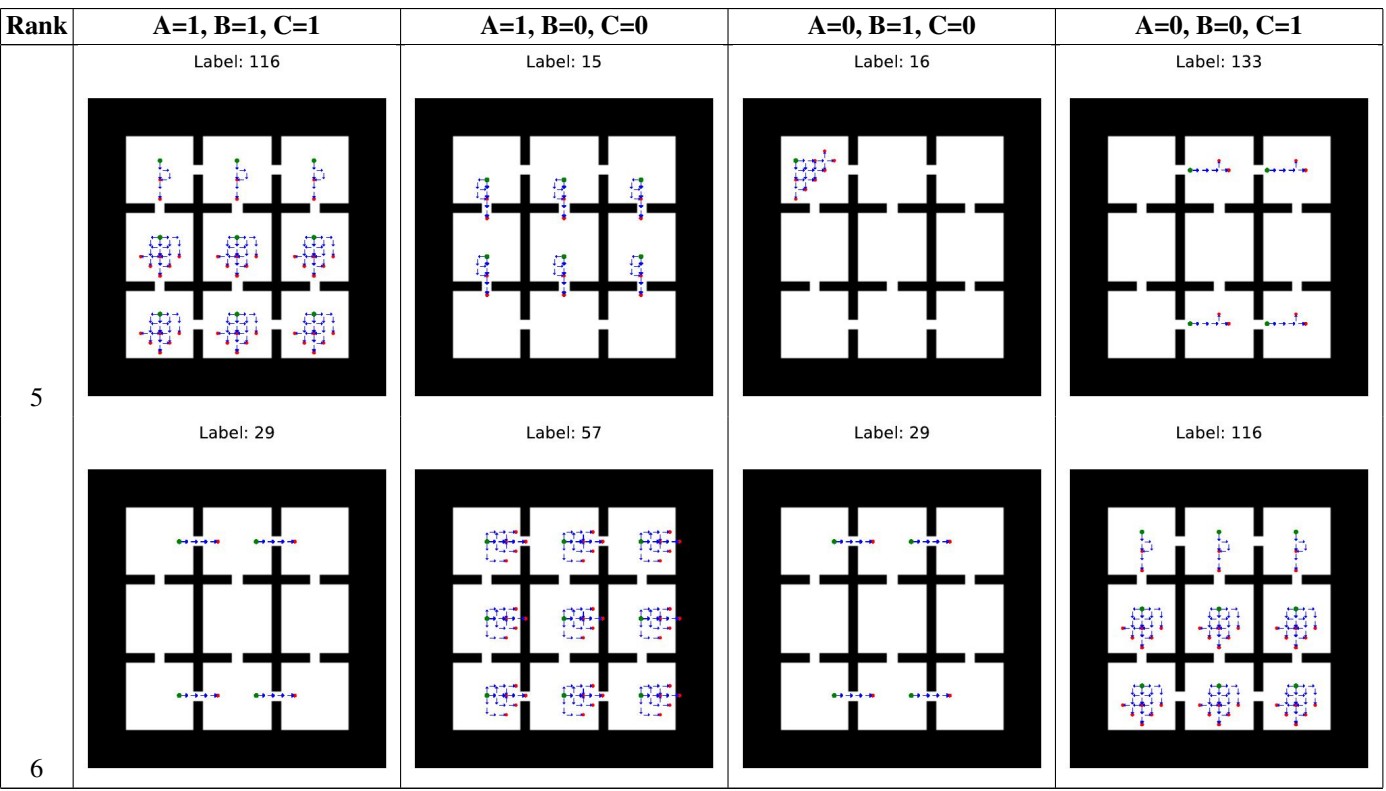

| Rank | A=1, B=1, C=1 | A=1, B=0, C=0 | A=0, B=1, C=0 | A=0, B=0, C=1 |
|------|---------------|---------------|---------------|---------------|
| 7 | Label: 35 | Label: 22 | Label: 58 | Label: 35 |
| 8 | Label: 133 | Label: 53 | Label: 133 | Label: 29 |

Table 12: Ranking of fracture clusters based on different weight configurations. Each cell contains an image visualisation of that fracture cluster. Here negative indicates worst rank. -1 is the lowest rank cluster for instance.

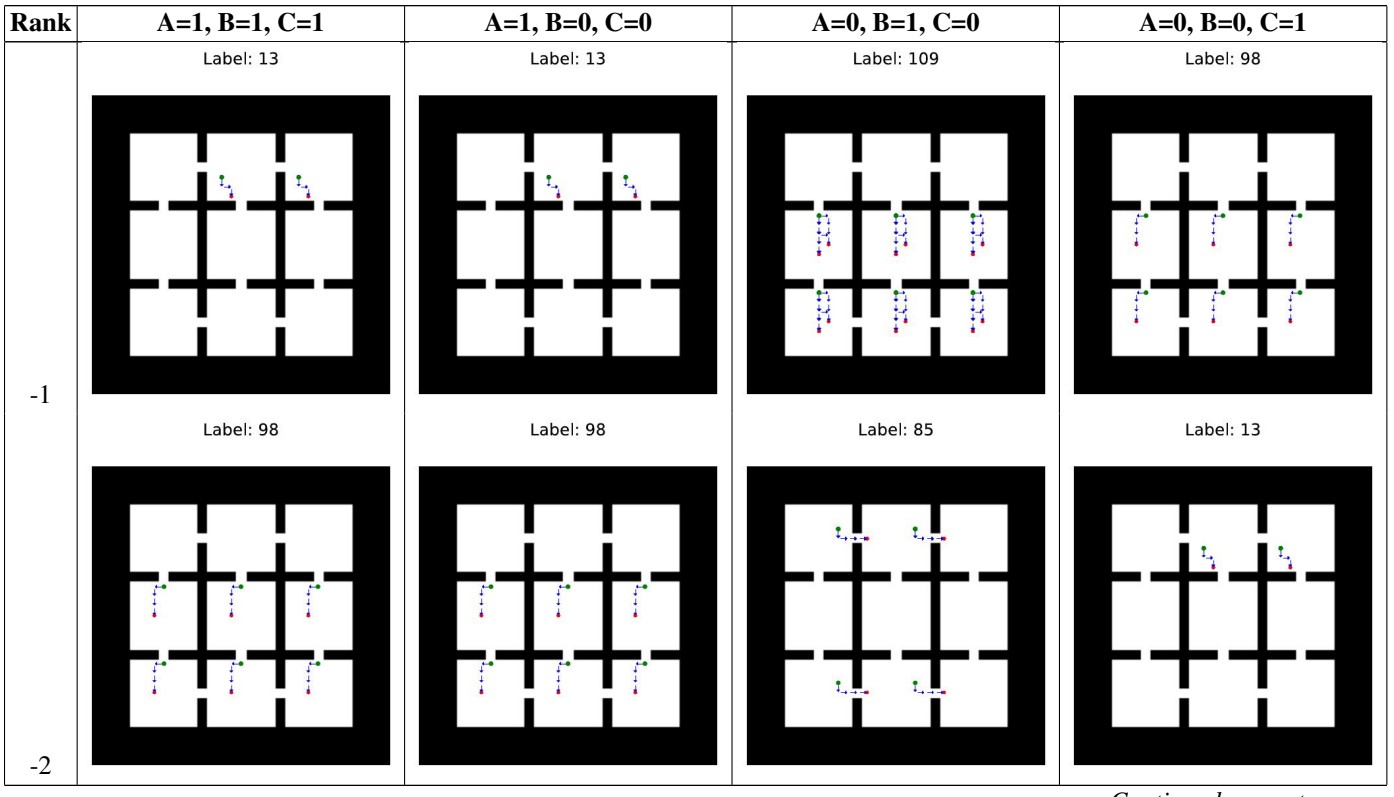

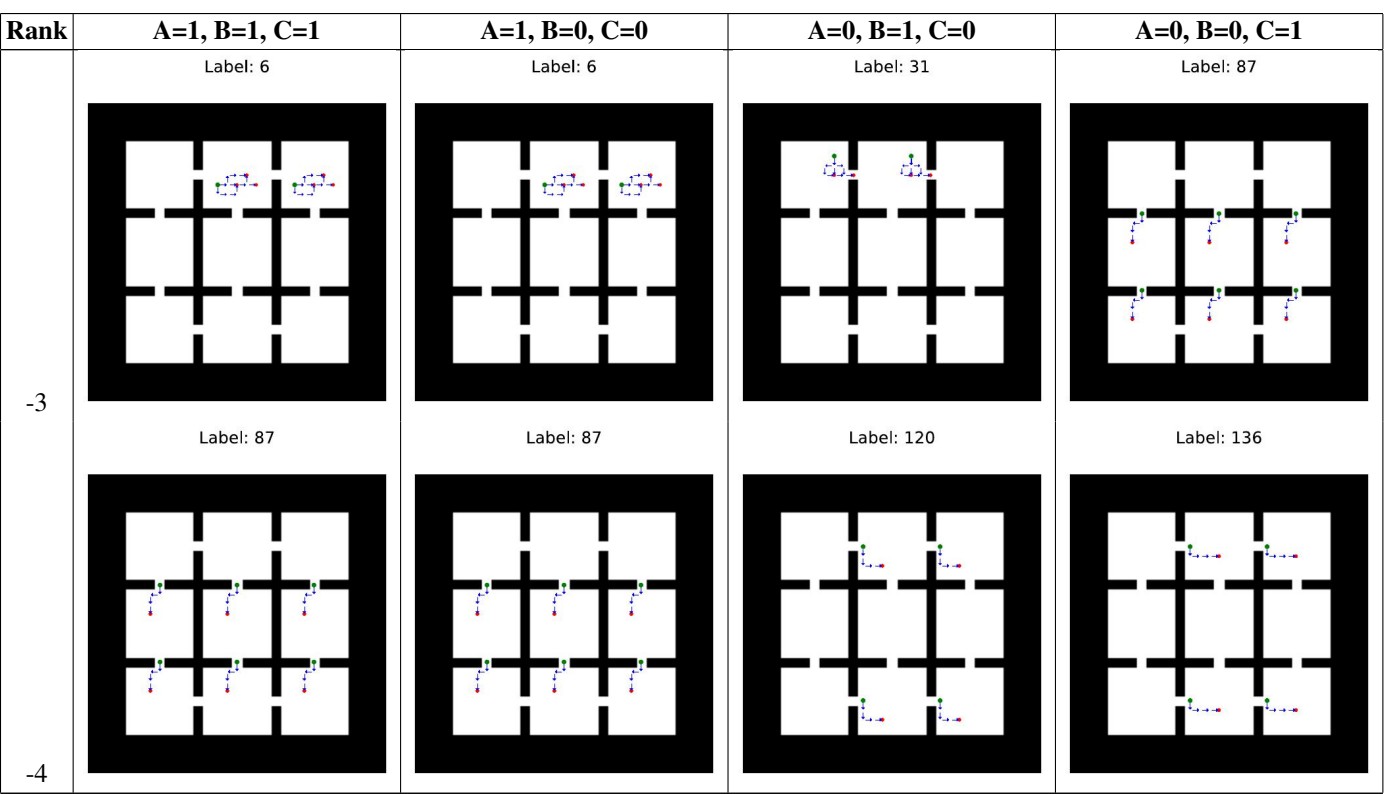

*Continued on next page...*

| Rank | A=1, B=1, C=1 | A=1, B=0, C=0 | A=0, B=1, C=0 | A=0, B=0, C=1 |
|---|---|---|---|---|

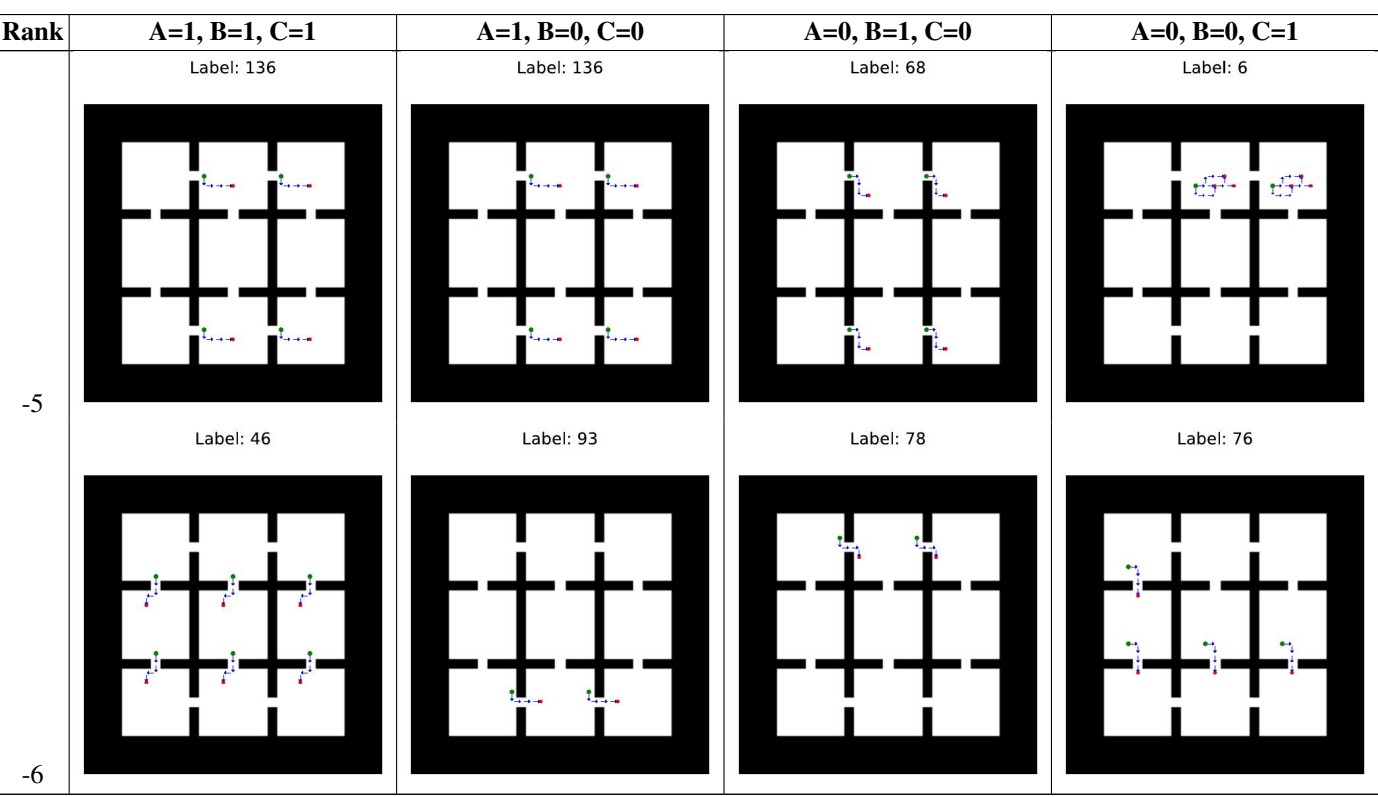

| Rank | A=1, B=1, C=1 | A=1, B=0, C=0 | A=0, B=1, C=0 | A=0, B=0, C=1 |
|------|---------------|---------------|---------------|---------------|
| -7 | Label: 76 | Label: 83 | Label: 93 | Label: 46 |
| -8 | Label: 2 | Label: 31 | Label: 83 | Label: 2 |

