# OpenReview forum: "Accelerating Task Generalisation with Multi-Level Skill Hierarchies"
_ICLR.cc/2025/Conference — ICLR 2025 Poster_

### Official Review · Reviewer_6Ht3 · 2024-10-29

**Soundness:** 2
**Presentation:** 2
**Contribution:** 3
**Rating:** 6
**Confidence:** 3

**Summary:**

The paper introduces Fracture Cluster Options (FraCOs), a multi-level hierarchical reinforcement learning method designed to improve generalization in new tasks. By identifying patterns in agent behavior and forming options based on expected usefulness, FraCOs achieves outperformance in both in-distribution and out-of-distribution settings. The approach demonstrates generalization and performance over existing deep reinforcement learning algorithms across complex procedurally generated environments.

**Strengths:**

•	Originality: The concept of fractures to find recurring patterns is innovative for hierarchical reinforcement learning. This paper proposes a method to cluster and select fractures, as well as a framework to utilize fractures in hierarchical reinforcement learning.

•	Quality: The paper effectively uses descriptive language and figures to illustrate core concepts.

•	Significance: The proposed method is shown to outperform several baselines.

**Weaknesses:**

1.	The Grid-world environments used in the experiments are limited in variety where all tasks are navigation across rooms. While Procgen environments were also tested, some details are lacking, such as the result of each environment, and the result of FraCOs in Figure 6 (also see question 5). In addition, I suggest that more baselines work on Procgen should also be compared (e.g. PPG[1]).

2.	In Section 4.1 and Figure 2, fractures are visualized with distinct clusters in 2D. I wonder if clusters in other tasks are as clearly separated as in this example, especially since the clustering approach is unsupervised and does not consider semantics.

3.	Based on the concern in question 2, I wonder about the reliability of using UMAP and HDBSCAN for clustering fractures. In Section 4.2, you state that fracture clusters are based on behavioral similarity, and you use an unsupervised approach to identify them. However,  Section 4.1 has provided a mathematical definition of fractures. It is unclear whether the identified clusters consistently align with this definition. Do you compare it with other clustering methods? Please clarify the methods further and provide additional analysis of the clustering results.

4.	The definition of usefulness (normalized sum) lacks explanation, with no supporting experiments or rationale for this decision. Would the proportion of each factor affect the result? Is there any ablation study to demonstrate the impact of each component?

5.	The FraCOs results without SSR are unconvincing, and Figure 6 does not even include FraCOs without SSR. SSR assumes shared convolutional layers encode the state, which is a strong assumption. The results would be more convincing with a clearer explanation of why SSR are applied to FraCOs in this part, including assumption and an analysis of its impact on the results. Providing the complete result on Procgen would also be helpful.

[1] Cobbe, Karl W., Jacob Hilton, Oleg Klimov, and John Schulman. "Phasic policy gradient." In International Conference on Machine Learning, pp. 2020-2027. PMLR, 2021.

**Questions:**

See above.

---

> ### Author Response · Authors · 2024-11-19
>
> We sincerely thank Reviewer 6Ht3 for their detailed feedback. We believe that we have addressed the weaknesses that were highlighted and therefore this feedback has directly improved the quality of our work.
>
> ### Summary of changes
>
> - Results of **all Procgen** environments are **now included** in the main text. All learning curves are also provided in Appendix A. 14
> - PPG has been compared.
> - Full FraCOs and FraCOs-SSR results are reported in the main body (Figures 6 and 7).
>     - FraCOs-SSR remains as the top performing method in IID and OOD tasks, FraCOs outperforms all other methods.
> - We have provided detail on the clustering approaches including sweeps of HDBSCAN hyperparameters in Appendix A. 16
> - An **ablation study** of the Expected Usefulness weightings has been added to Appendix A. 17

---

> ### Author Response · Authors · 2024-11-19
>
> ### Detailed responses
>
> ### Response to Questions 2 and 3
>
> > 2. In Section 4.1 and Figure 2, fractures are visualized with distinct clusters in 2D. I wonder if clusters in other tasks are as clearly separated as in this example, especially since the clustering approach is unsupervised and does not consider semantics.
> >
>
> > 3. Based on the concern in question 2, I wonder about the reliability of using UMAP and HDBSCAN for clustering fractures. In Section 4.2, you state that fracture clusters are based on behavioral similarity, and you use an unsupervised approach to identify them. However, Section 4.1 has provided a mathematical definition of fractures. It is unclear whether the identified clusters consistently align with this definition. Do you compare it with other clustering methods? Please clarify the methods further and provide additional analysis of the clustering results.
> >
>
> ---
>
> **Clarification on UMAP Usage**
>
> We first clarify that **UMAP is used only for visualisation purposes** in this work and is not part of the clustering process. Fractures are not reduced in dimensionality using UMAP during experiments. Instead, HDBSCAN operates on the full feature space of the fractures, ensuring that the clustering process aligns directly with the data representation.
>
> ---
>
> **Challenges in Procgen and Our Approach**
>
> We encountered challenges using HDBSCAN in the **Procgen environments** due to the large state-space dimensionality (64 × 64 × 3). It is well-documented that HDBSCAN struggles with performance beyond 100 features. To address this, we trained a CNN-VAE to reduce the dimensionality of the state space. While this approach produced mostly sensible clusters, the occasional unreliability of the clustering introduced instability during training.
>
> Given these challenges, we opted to simplify the clustering for Procgen by grouping fractures based on **action sequences alone**. We then trained a neural network (NN) to predict cluster membership for initiation states, effectively treating the NN as a state classifier for each cluster. This hybrid approach proved to be a reliable technique for handling the complexity of Procgen environments and maintained the integrity of the FraCOs framework.
>
> ---
>
> The **FraCOs framework is agnostic to the clustering mechanism**. The mathematical definition of fracture usefulness, particularly P(ϕ∈z)P(\phi \in z)P(ϕ∈z), represents the predicted probability that fracture ϕ\phiϕ belongs to cluster zzz. This probability can be derived from any clustering method, whether through direct prediction (as in HDBSCAN) or approximation (as with nearest neighbors) or with Neural Network predictions (as we do in the Procgen experiments). We do agree that the choice of clustering does impact the quality of the learned options and should be considered carefully in implementation.
>
> ---
>
> **Analysis of Clustering Methods**
>
> We analysed various clustering methods to evaluate their suitability for our framework. HDBSCAN was chosen for its ability to automatically determine the number of clusters and to provide direct prediction on new data points. This contrasts with methods like DBSCAN, Mean Shift, and Optics, which also do not require a predefined number of clusters but lack built-in prediction mechanisms. For these methods, we implemented nearest-neighbour prediction for cluster assignment on new data but this did require storing all clustering data.
>
> We also tested methods that require a predefined number of clusters, such as K-Means and Gaussian Mixture Models (GMMs), across a range of cluster numbers.
>
> Below, we summarise the clustering methods tested and provide the visual examples of the clustering in Appendix A.16
>
> | Method | Prediction Mechanism | Predetermined clusters? | Hyperparameters checked |
> | --- | --- | --- | --- |
> | HDBSCAN  | Built-in | No | min_cluster_size, metric |
> | DBSCAN [1] | Nearest-neighbour | No | eps, min_samples |
> | Mean Shift [2] | Nearest-neighbour | No | bandwidth |
> | Optics [3] | Nearest-neighbour | No | min_samples |
> | K-Means  | Built-in (centroid) | Yes | n_clusters |
> | GMM  | Built-in (soft probability | Yes | n_components |
>
> HDBSCAN, DBSCAN and Mean Shift all performed well, however we decided on HDBSCAN with a Euclidean distance metric and min_cluster_size of 15. This was chosen due to performance and simplicity of handling prediction on new data-points. This was used for all clustering in our tabular experiments.

---

> ### Author Response · Authors · 2024-11-19
>
> **Expected Usefulness Weights.**
>
> > 4. The definition of usefulness (normalized sum) lacks explanation, with no supporting experiments or rationale for this decision. Would the proportion of each factor affect the result? Is there any ablation study to demonstrate the impact of each component?
> >
>
> We now provide an ablation study in Appendix A.17, conducted in the Nine Rooms environment, by introducing hyperparameters $A, B, C$ to weight each term in the Expected Usefulness formula:
>
> $$
> E[U_{(\phi_c)}] = \frac{1}{3} \left(A(\text{Appearance Probability}) + B(\text{Relative Frequency}) - C(\text{Estimate Entropy}) \right).
> $$
>
> We visualise the top and bottom eight fracture clusters ranked by combinations of A, B and C. These visualisations are provided in Appendix A. 17, a summary of the observations are provided here:
>
> | A | B | C | Observations |
> | --- | --- | --- | --- |
> | 1 | 1 | 1 | Equal weights highlight diverse fracture clusters, balancing appearance, frequency and entropy. |
> | 0 | 0 | 1 | Prioritises intuitively useful fracture clusters but specific trends are unclear  |
> | 0 | 1 | 0 | Prioritises frequent occurring fractures often near the initiation state. |
> | 1 | 0 | 0 | Prioritises fractures common in successful trajectories. Less frequently chooses fractures near the initial state. More often chooses bottlenecks. |
>
> This ablation confirms that the choice of weights does impact fracture cluster selection and this may be leveraged for specific applications. However, we believe that the current equal-weight assumption is a reasonable starting point for the base algorithm. Furthermore it is not without *some* precedent see [4], [5].
>
> > 5. The FraCOs results without SSR are unconvincing, and Figure 6 does not even include FraCOs without SSR. SSR assumes shared convolutional layers encode the state, which is a strong assumption. The results would be more convincing with a clearer explanation of why SSRs are applied to FraCOs in this part, including assumptions and an analysis of their impact on the results. Providing the complete result on Procgen would also be helpful.
> >
>
> We appreciate your feedback on this important point. In response, we have made several updates to address these concerns:
>
> 1. **Expanded Benchmark Analysis**: We now include results for **FraCOs**, **FraCOs-SSR**, **OC-PPO**, **PPG**, and **PPO** in our benchmark analysis. Additionally, the full set of results on Procgen environments is now presented in the main body of the paper in **Figures 6 and 7**.
> 2. **Context for SSRs in FraCOs and OC-PPO**: Shared State Representations (SSRs), implemented as a shared series of convolutional layers, are used to encode the state space. These layers feed into task-specific linear layers (e.g., actor, critic, or option modules). While FraCOs does not require the SSR to produce meaningful options, OC-PPO relies heavily on the SSR, as the shared representation directly feeds into the option policy. Resetting the meta-policy entirely (including the SSR) during transfer tasks can destabilise OC-PPO, making it difficult to maintain a meaningful set of options.
>
>     To ensure a fair comparison between FraCOs and OC-PPO, we retained the SSR in FraCOs during training on unseen tasks. This setup allowed for a more consistent baseline across methods and avoided advantages unrelated to the core methodology. Importantly, even without SSR, **FraCOs consistently outperformed OC-PPO** in our experiments.
>
> 3. **Detailed Algorithmic Assumptions**: We recognise that applying SSRs introduces assumptions about shared feature representations across tasks. We have expanded the discussion of these assumptions and their implications in the main text and provided additional details on algorithm architectures, including SSRs, in Appendix A.11.
> 4. **Impact Analysis of SSRs**: To further address this concern, we have clarified the role of SSRs in FraCOs and analyzed their impact on the results. The inclusion of SSRs in FraCOs is primarily to align with OC-PPO for fair comparison, rather than being an intrinsic requirement of FraCOs. This distinction is now made explicit in the revised paper.
>
> [1] Ester, Martin, et al. "A density-based algorithm for discovering clusters in large spatial databases with noise." *kdd*. Vol. 96. No. 34. 1996.
>
> [2] Comaniciu, Dorin, and Peter Meer. "Mean shift: A robust approach toward feature space analysis." *IEEE Transactions on pattern analysis and machine intelligence* 24.5 (2002): 603-619.
>
> [3] Ankerst, Mihael, et al. "OPTICS: Ordering points to identify the clustering structure." *ACM Sigmod record* 28.2 (1999): 49-60.
>
> [4] Eysenbach, Benjamin, et al. "Diversity is all you need: Learning skills without a reward function." arXiv preprint arXiv:1802.06070 (2018).
>
> [5] Lichtenberg, Jan Malte, and Özgür Şimşek. "Regularization in directable environments with application to Tetris." International Conference on Machine Learning. PMLR, 2019.

---

> ### Comment · Reviewer_6Ht3 · 2024-11-22
> **Official Comment by Reviewer 6Ht3**
>
> Thank you for your response to my comments. The work is more convincing with these explanations and updates. I have updated my score accordingly.

---

### Official Review · Reviewer_Cp7c · 2024-11-04

**Soundness:** 3
**Presentation:** 3
**Contribution:** 3
**Rating:** 8
**Confidence:** 3

**Summary:**

This paper introduces a method for learning options which can be used to generalize across tasks.
FraCOs are learned based on patterns of a state and multiple subsequent actions observed in trajectories which reach a desired return threshold and are thus deemed successful.
The authors present results from multiple environments which demonstrate how the use of FraCOs improves learning across tasks which may vary in state space or reward function.

**Strengths:**

The problem of learning options (or other forms of sub-tasks) is an important issue in hierarchical RL.

The paper is well-written and generally easy to follow.

The motivation for the method provided in Section 4 is strong.

The experimental results effectively demonstrate the benefit of FraCOs in terms of generalizing between tasks and accelerating learning in new tasks in discrete state spaces.

**Weaknesses:**

There is a lack of discussion of continuous state spaces and no experiment(s) involving them.

It seems that a lot of data is required to learn the FraCOs before learning the actual policy, e.g., allowed to discover FraCOs in 50 of the 60 tasks used in Experiment 1. It would be nice to see a comparison of how the amount of FraCO pre-training affects the learning in later tasks and how that relates to, e.g., OC-PPO.

**Questions:**

Is there a reason that no continuous state spaces are used?

While the components of the usefulness metric all seem reasonable to include, is there any justification (possibly experimental) for why they are weighted the same?

---

> ### Author Response · Authors · 2024-11-19
>
> We sincerely thank Reviewer Cp7c for their thoughtful and positive feedback. We are especially grateful for your recognition of the importance of learning options in hierarchical RL and for highlighting the strength of our method’s motivation and experimental results. Your comments on the soundness and clarity of our work are deeply appreciated.
>
> **Continuous state spaces**
>
> > Is there a reason that no continuous state spaces are used?
> >
>
> Thank you for raising this important point. While we have not explicitly tested FraCOs in continuous state spaces, we note that Procgen environments, while technically discrete, exhibit a near-infinite discrete state space. With 64x64x3 features, each ranging from 0–255, the combinatorial complexity approximates many challenges found in continuous state spaces.
>
> Testing FraCOs in environments specifically designed for continuous state and *action spaces,* such as Meta-World [1], is a natural next step. While our current work focuses on discrete action spaces to clearly explain the core principles of FraCOs, we believe the method can be adapted for continuous settings with the following approach:
>
> 1. **Bucketing actions**: Discretising continuous actions into bins to form a structured basis for fractures.
> 2. **Forming fractures**: Leveraging these bucketed actions to define fractures.
> 3. **Clustering fractures**: Applying the FraCO framework to group fractures based on behavioural similarity.
> 4. **Prediction and policy learning**: Using neural networks to predict initiation states and to learn FraCO policies, as we demonstrated in Procgen experiments.
>
> By exploring environments like Meta-World, FraCOs could address generalisation challenges across a broader range of real-world settings (such as robotics). While we chose not to include these adaptations in this submission to maintain clarity and focus, we see this as a promising direction for future work.
>
> **Expected Usefulness Weights.**
>
> > While the components of the usefulness metric all seem reasonable to include, is there any justification (possibly experimental) for why they are weighted the same?
> >
>
> We now provide an ablation study in Appendix A. 17, conducted in the Nine Rooms environment, by introducing hyperparameters $A, B, C$ to weight each term in the Expected Usefulness formula:
>
> $$
> E[U_{(\phi_c)}] = \frac{1}{3} \left(A(\text{Appearance Probability}) + B(\text{Relative Frequency}) - C(\text{Estimate Entropy}) \right).
> $$
>
> We visualise the top and bottom eight fracture clusters ranked by combinations of A, B and C. These visualisations are provided in Appendix A. 17, a summary of the observations are provided here:
>
> | A | B | C | Observations |
> | --- | --- | --- | --- |
> | 1 | 1 | 1 | Equal weights highlight diverse fracture clusters, balancing appearance, frequency and entropy. |
> | 0 | 0 | 1 | Prioritises intuitively useful fracture clusters but specific trends are unclear  |
> | 0 | 1 | 0 | Prioritises frequent occurring fractures often near the initiation state. |
> | 1 | 0 | 0 | Prioritises fractures common in successful trajectories. Less frequently chooses fractures near the initial state. More often chooses bottlenecks. |
>
> This ablation confirms that the choice of weights does impact fracture cluster selection and this may be leveraged for specific applications. However, we believe that the current equal-weight assumption is a reasonable starting point for the base algorithm. Furthermore it is not without *some* precedent see [3], [4].
>
> [1] Yu, Tianhe, et al. "Meta-world: A benchmark and evaluation for multi-task and meta reinforcement learning." Conference on robot learning. PMLR, 2020.
>
> [2]  Eysenbach, Benjamin, et al. "Diversity is all you need: Learning skills without a reward function." arXiv preprint arXiv:1802.06070 (2018).
>
> [3] Lichtenberg, Jan Malte, and Özgür Şimşek. "Regularization in directable environments with application to Tetris." International Conference on Machine Learning. PMLR, 2019.

---

> > ### Comment · Reviewer_Cp7c · 2024-12-03
> >
> > I appreciate the authors' responses.
> >
> > Can you say anything about the amount of data required to learn the FraCOs? I mentioned this in the weaknesses, though didn't ask a direct question on it.
> >
> > Regarding the lack of continuous state spaces: it is worth noting that while the image-based observation space is large, the underlying state space is drastically smaller. The vast majority of observations have no corresponding state from which they would be generated. I think it is important to address this in the text, even if you don't have an experiment with a continuous state space.
> >
> > Thank you for the additional information regarding the usefulness metric.

---

> > > ### Author Response · Authors · 2024-12-03
> > >
> > > We thank reviewer Cp7c for the opportunity for further discussion. They raise some interesting questions.
> > >
> > > **FraCOs Data Requirements**
> > >
> > > We acknowledge the importance of understanding data requirements for FraCOs and outline potential investigations below. While not included in the main text to maintain clarity, we plan to explore these in our code release and camera-ready supplementary materials.
> > >
> > > **Task Availability and Quality**:
> > >
> > > - In our tabular experiments, we sample ~80% of all possible tasks, while in Procgen, we use 32 tasks, representing 50% of available tasks. However, each task in Procgen is more complex with longer trajectories.
> > > - We believe that experiments comparing the percentage of available tasks and the impact on FraCOs performance would be insightful. Intuitively, more tasks should improve performance but may reduce the considered generalisation performance.
> > > - This relationship is likely environment-specific; high variability tasks may require more samples for more effective FraCOs. However, even with fewer tasks, FraCOs should not degrade performance below baseline due to sensible initiation conditions.
> > > - In application we suggest developing FraCOs on the maximum available tasks, but agree that identifying a threshold for effective FraCOs development would be valuable knowledge.
> > >
> > > **Environment Interactions**:
> > >
> > > - FraCOs can be formed from demonstrations, which would require no direct environment interactions. This approach is feasible for single-level hierarchies, where human demonstrations can provide sufficient data for initial FraCO formation.
> > > - However, forming deeper levels of hierarchy presents challenges. Each subsequent level of FraCOs relies on the previous levels to define new, more abstract options. Demonstrations for these higher levels would need to incorporate the previously learned FraCOs, which are not typically available. We suspect that these could be formed by refactoring the original demonstration trajectories to include FraCOs. However, we decided that iteratively learning FraCOs from interactions was a more flexible approach.
> > > - With this in mind, we compare FraCOs and PPO with equal interactions, allowing FraCOs to maintain a shared state representation through iterations. Results show FraCOs improved out-of-distribution (OOD) performance while maintaining a high in-distribution (IID) performance.
> > >
> > > |  | Final IID | Final OOD |
> > > | --- | --- | --- |
> > > | FraCOs-SSR | 0.923 | **0.617** |
> > > | PPO-25 | **1.000** | 0.489 |
> > >
> > > Table 1. Normalised mean performances of all tested Procgen environments after 25 million total interactions.
> > >
> > > ---
> > >
> > > > Regarding the lack of continuous state spaces: it is worth noting that while the image-based observation space is large, the underlying state space is drastically smaller. The vast majority of observations have no corresponding state from which they would be generated. I think it is important to address this in the text, even if you don't have an experiment with a continuous state space.
> > > >
> > >
> > > This is a valid point we hadn't fully considered. We will include a discussion on continuous state spaces in the camera-ready version. Thank you for highlighting this.
> > >
> > > We hope this addresses your questions. Thank you again for helping us improve our paper and for the positive feedback.

---

### Official Review · Reviewer_R928 · 2024-11-04

**Soundness:** 3
**Presentation:** 2
**Contribution:** 3
**Rating:** 6
**Confidence:** 4

**Summary:**

This paper focuses on the problem of reinforcement learning policies having a limited ability to generalize or adapt quickly to new environments. In order to overcome this limitation, the authors propose Fracture Cluster Options (FraCO). FraCOs leverage the history of trajectories on a set of training tasks, to create a set of options (alongside the primitive actions of the environment) for the agent to use during the testing phase where the agent is trained on the test tasks. The fracture is defined as the induced trajectory from state $s_t$, using actions $a_t, a_{t+1}, a_{t+2}, ..., a_{t+b-1}$ up to some length $b$. The motivation for doing this is given through a visualization of the latent space of the four rooms environment in which clusters of fractures show similar behaviours. In order to determine what options should be learned in this post-training phase, the authors create an Expected Usefulness metric based on the (1) Appearance Probability of a fracture in a trajectory, (2) Relative Frequency that some fracture appears in a successful trajectory, and (3) Entropy of Usage of that fracture.

The authors then demonstrate the performance of their FraCO method on 4 standard grid-world environments, Metagrid environments, and Procgen environments.

**Strengths:**

1. Novel approach for discovering re-usable options to accelerate generalization abilities
2. Thorough empirical evaluation of the FraCO method across several benchmarks
3. Clear & honest evaluation of how well the FraCO method may work going forward to additional environments/settings

**Weaknesses:**

1. The paper is written in a manner to accelerate the generalization abilities of RL agents. However, there isn't a single mention of sample complexity in the paper. All of the plots and figures are concerned with the overall success rate of the agent.
2. The given results across the 3 benchmarks are difficult to understand given that they are all success rate plots. In reinforcement learning, it is much better practice to use IQM plots [1] in order to compare the performance of multiple algorithms/configurations.
3. I strongly disagree with the decision to not tune the hyperparameters of the PPO algorithm when adding the additional option methods. It is very difficult to argue that PPO + options with the same hyperparameters as PPO alone. The hyperparameters of each method should have been tuned to ensure that a fair comparison has been made.
4. The organization of the paper could be vastly improved. It was very difficult to find relevant formulas/definitions/etc in the current version because they are mixed in with the motivating example(s). I think it would benefit the overall quality of the writing to separate these two components.
5. The final experiments in the Procgen environments should be run for more random seeds. I would say that 5 is a minimum, 10 is ideal.
6. The weights of the Expected Usefulness should be empirically chosen. In addition, each component should be ablated (i.e. only use component 1) in order to determine what each of the 3 components of the metric add to the framework.



[1] https://arxiv.org/abs/2108.13264

**Questions:**

1. In the Procgen experiments the authors disregarded the state information and only clustered based on the action sequences. Did you experiment with the addition of state information and find that it was causing issues (i.e. it was too noisy to use) ?
2. Could FraCO be used to facilitate better exploration of the state space? Or any other goals of learning options?
3. The use of a GAN to augment the dataset of Fracture Clusters was really interesting. Could this also be used in other regimes (i.e. a fixed computational budget regime) to learn FraCOs (or options in general) ?
4. What kinds of skills/actions do the FraCOs learn?

---

> ### Author Response · Authors · 2024-11-19
>
> We sincerely thank Reviewer R928 for their detailed feedback. It has significantly improved the quality of our work. In response, we have addressed the identified weaknesses and provided additional analyses to strengthen our contribution. Below, we summarize the major updates and provide detailed responses to each comment.
>
> ### Summary of Changes:
>
> - Procgen experiments have been re-run with **eight seeds** for statistical reliability.
> - All plots now use **Interquartile Mean (IQM)**, as recommended, for fair comparisons.
> - A **hyperparameter search** was conducted for OC-PPO, with new results added to the main text and Appendix A.17.
> - An **ablation study** of the Expected Usefulness weightings has been included.
> - A **glossary of terms** has been added to the start of the appendix for improved clarity and navigation.
>
> ### Detailed Responses:
>
> **Hyperparameter tuning.**
>
> > 3. I strongly disagree with the decision to not tune the hyperparameters of the PPO algorithm when adding the additional option methods. It is very difficult to argue that PPO + options with the same hyperparameters as PPO alone. The hyperparameters of each method should have been tuned to ensure that a fair comparison has been made.
> >
>
> To address this, we conducted a search over two key OC-PPO hyperparameters—**learning rate** and **entropy coefficient**—on a subset of Procgen environments. This search included 105 experiments, each running for 2M steps. The best-performing settings (learning rate = 0.001, entropy coefficient = 0.001) have been re-run for eight seeds and are now the results in the main body. Previously, these values were 0.0005 and 0.01. Below is a table which summarises the outcome of the sweep.
>
> | learning rate | entropy coefficient | Final Returns |
> | --- | --- | --- |
> | 0.00001 |  | **0.83** |
> |  | 0.001 | 0.79 |
> |  | 0.1 | 0.86 |
> | 0.0001 |  | **3.2** |
> |  | 0.001 | 5.67 |
> |  | 0.01 | 1.92 |
> |  | 0.1 | 0.83 |
> | 0.001 |  | **12.67** |
> |  | 0.001 | **15.83** |
> |  | 0.01 | 12.22 |
> |  | 0.1 | 11.75 |
> | Grand Total |  | 5.71 |
>
> **Note:** Hyperparameters for FraCOs remain untuned to isolate the effects of the method on PPO.
>
> **Expected Usefulness Weights.**
>
> > 6. The weights of the Expected Usefulness should be empirically chosen. In addition, each component should be ablated (i.e. only use component 1) in order to determine what each of the 3 components of the metric add to the framework.
> >
>
> We now provide an ablation study in Appendix A.17, conducted in the Nine Rooms environment, by introducing hyperparameters $A, B, C$ to weight each term in the Expected Usefulness formula:
>
> $$
> E[U_{(\phi_c)}] = \frac{1}{3} \left(A(\text{Appearance Probability}) + B(\text{Relative Frequency}) - C(\text{Estimate Entropy}) \right).
> $$
>
> We visualise the top and bottom eight fracture clusters ranked by combinations of A, B and C. These visualisations are provided in Appendix A.17, a summary of the observations are provided here:
>
> | A | B | C | Observations |
> | --- | --- | --- | --- |
> | 1 | 1 | 1 | Equal weights highlight diverse fracture clusters, balancing appearance, frequency and entropy. |
> | 0 | 0 | 1 | Prioritises intuitively useful fracture clusters but specific trends are unclear  |
> | 0 | 1 | 0 | Prioritises frequent occurring fractures often near the initiation state. |
> | 1 | 0 | 0 | Prioritises fractures common in successful trajectories. Less frequently chooses fractures near the initial state. More often chooses bottlenecks. |
>
> This ablation confirms that the choice of weights does impact fracture cluster selection and this may be leveraged for specific applications. However, we believe that the current equal-weight assumption is a reasonable starting point for the base algorithm. Furthermore it is not without *some* precedent see [1], [2].

---

> ### Author Response · Authors · 2024-11-19
>
> **Paper organisation.**
>
> > 4. The organization of the paper could be vastly improved. It was very difficult to find relevant formulas/definitions/etc in the current version because they are mixed in with the motivating example(s). I think it would benefit the overall quality of the writing to separate these two components.
> >
>
> We acknowledge the difficulty in balancing definitions and motivating examples. While some reviewers appreciated the current structure, we understand your concerns. To improve clarity we have now included a glossary of terms at the start of the appendix.
>
> >
> >
> > 1. The paper is written in a manner to accelerate the generalization abilities of RL agents. However, there isn't a single mention of sample complexity in the paper. All of the plots and figures are concerned with the overall success rate of the agent.
> > 2. The given results across the 3 benchmarks are difficult to understand given that they are all success rate plots. In reinforcement learning, it is much better practice to use IQM plots [1] in order to compare the performance of multiple algorithms/configurations.
>
> We understand how the structure of our plots, with a final reward of +1 and low action cost of -0.01, could lead to the interpretation that they are success rate plots. However, we intended them to be learning curves that demonstrate the FraCOs agent’s ability to achieve higher returns in fewer environment interactions. However, we agree that Interquartile Mean (IQM) plots are more valid here since we are averaging over 10 different tasks. We have made these adjustments as the reviewer has recommended.
>
> Regarding sample complexity, we appreciate the importance of explicitly addressing this point. While our results implicitly showcase sample efficiency—for instance, in the grid-world experiments (Figures 4 and 5), where deeper hierarchies enable agents to learn optimal solutions in fewer steps—we recognise that this was not clearly articulated in the original text. We have made adjustments to the writing.
>
> [1] Eysenbach, Benjamin, et al. "Diversity is all you need: Learning skills without a reward function." arXiv preprint arXiv:1802.06070 (2018).
>
> [2] Lichtenberg, Jan Malte, and Özgür Şimşek. "Regularization in directable environments with application to Tetris." International Conference on Machine Learning. PMLR, 2019.

---

> ### Author Response · Authors · 2024-11-19
>
> ### Questions:
>
> > 1. In the Procgen experiments the authors disregarded the state information and only clustered based on the action sequences. Did you experiment with the addition of state information and find that it was causing issues (i.e. it was too noisy to use) ?
> >
>
> Thank you for this insightful question. We did experiment with incorporating both state and action information into the cluster formations for Procgen. However, as the state space grew larger, the clustering predictions became less reliable. This aligns with known limitations of HDBSCAN, which struggles with performance when feature counts exceed 100.
>
> To address this, we attempted dimension reduction techniques, such as a CNN-VAE. This approach produced some reasonable options, not all were, leading to performance comparable to PPO (the base algorithm).
>
> After testing several clustering methods, we found that clustering based solely on action sequences, combined with a neural network to predict initiation states, was both more efficient and reliable for the deep experiments.
>
> > 3. The use of a GAN to augment the dataset of Fracture Clusters was really interesting. Could this also be used in other regimes (i.e. a fixed computational budget regime) to learn FraCOs (or options in general) ?
> >
>
> This is an excellent follow-up to the previous question. In our work, we formed Fracture Clusters from a relatively small number of trajectories (around 100), resulting in approximately 10,000 state-action pairs. Since clustering often produced many small clusters, some had as few as 50 samples, making it difficult to train a neural network to predict initiation states effectively.
>
> To address this, we used a GAN to up-sample the dataset, improving the reliability of our initiation state predictions. This approach could certainly be adapted for fixed computational budget regimes or other data-scarce scenarios. While we are not aware of other methods that directly use GANs for this purpose, there is related work that uses GANs to augment reinforcement learning trajectories, such as [3]. This connection highlights a potential avenue for further exploration.
>
> We conducted a study of GAN produced state quality. However, we felt this was out of scope to include in this project.
>
> > 2. Could FraCO be used to facilitate better exploration of the state space? Or any other goals of learning options?
> >
>
> Hierarchical RL methods, such as options, are known to improve exploration by enabling temporally extended actions, guiding the agent toward meaningful transitions, and biasing behaviour toward promising areas of the state space [4, 5, 6].
>
> FraCO’s fracture clusters, which capture re-usable sub-trajectories, likely also guide exploration by focusing on high-value or diverse behaviours. However, validating this effect is currently outside the scope of this work. Future research could explore FraCO’s potential as a tool for improving structured exploration, particularly in environments with sparse rewards or complex state spaces.
>
> > 4. What kinds of skills/actions do the FraCOs learn?
> >
>
> In Figures 2 and 3, we visualise options discovered at a single level of hierarchy in simple grid-world environments. These options predominantly identify bottlenecks or guide the agent away from the initial state, which are intuitively meaningful behaviours. FraCOs operate bottom-up, with deeper options composed of sequentially linked lower-level options, where the termination state of one aligns with the initiation state of the next.
>
> In Procgen the discovered options exhibited sensible behaviours, such as moving upward or to the right in platforming environments (e.g., Coinrun, Ninja) or avoiding and interacting with objects in more complex tasks (e.g., Bigfish, Plunder, Starpilot).
>
> We plan to provide more detailed examples and visualisations of Procgen skills in open-source blogs upon publication.
>
> [1]  Eysenbach, Benjamin, et al. "Diversity is all you need: Learning skills without a reward function." arXiv preprint arXiv:1802.06070 (2018).
>
> [2] Lichtenberg, Jan Malte, and Özgür Şimşek. "Regularization in directable environments with application to Tetris." International Conference on Machine Learning. PMLR, 2019.
>
> [3] Janner, Michael, et al. "Planning with diffusion for flexible behavior synthesis." arXiv preprint arXiv:2205.09991 (2022).
>
> [4] Stolle, M., & Precup, D. (2002). "Learning options in reinforcement learning." *Advances in Neural Information Processing Systems (NIPS)*.
>
> [5] Nachum, O., Gu, S. S., Lee, H., & Levine, S. (2018). "Data-efficient hierarchical reinforcement learning." *Advances in Neural Information Processing Systems (NeurIPS)*.
>
> [6] Kulkarni, T. D., Narasimhan, K., Saeedi, A., & Tenenbaum, J. B. (2016). "Hierarchical deep reinforcement learning: Integrating temporal abstraction and intrinsic motivation." *Advances in Neural Information Processing Systems (NIPS)*.

---

> > ### Comment · Reviewer_R928 · 2024-11-21
> >
> > Thank you for such a thorough review of my comments and update of your paper. Upon seeing these additions, I believe this makes the work much stronger. I have updated my score accordingly.

---

### Official Review · Reviewer_11zq · 2024-11-04

**Soundness:** 3
**Presentation:** 3
**Contribution:** 3
**Rating:** 6
**Confidence:** 3

**Summary:**

This work introduces fractures for learning bottom-up abstractions by clustering agent behavior across tasks.

**Strengths:**

* Paper is generally well-written.
* Related work appears exhaustive.
* Limitations are adequately addressed.
* Results are generally compelling.
* Findings (with potential caveats; see below) are likely to be useful to the broader field.

**Weaknesses:**

- The only domain where appropriate baselines are compared against is ProcGen. The results are still compelling, but another domain where baselines are also evaluated would have been useful to benchmark FraCO's relative utility. Additionally, some didactic experiments in the tabular settings with more appropriate baselines would be useful.
- Despite discussing more advanced baselines in the related work (most notably HOC, which can support multiple levels of hierarchy), OC is the only one used.

**Questions:**

- HOC is not used as a baseline; why not?
- FraCO is not compared against OC-PPO (or any other hierarchical approaches) in the tabular settings; why not?

---

> ### Author Response · Authors · 2024-11-19
>
> We sincerely thank Reviewer 11zq for their positive feedback and thoughtful suggestions. We are especially grateful for your recognition of the work's soundness, compelling results, and potential contributions to the broader field. Below, we address the questions and weaknesses raised in your review.
>
> **Hierarchical Option Critic.**
>
> > • HOC is not used as a baseline; why not?
> >
>
> We initially intended to use Hierarchical Option Critic (HOC) as the main benchmark method due to its support for multi-level hierarchies. However, we found HOC challenging to train in diverse task environments, as it frequently suffered from instability and option collapse. To address this, we conducted an extensive hyperparameter sweep, testing 170 configurations across learning rates, entropy coefficients, and the number of options at each hierarchy level (two levels). Unfortunately, only five combinations avoided option collapse, and none of these demonstrated significant learning progress across environments.
>
> We believe these results highlight the robustness and ease of FraCOs as a multi-level hierarchical method compared to HOC, which we found difficult to work with in practice. Below is a summary of our most successful HOC configurations:
>
> | Number of Meta Options | Number of Options | Learning Rate | Entropy Coefficient | Environment | Return (2 Million Steps) |
> | --- | --- | --- | --- | --- | --- |
> | 8 | 4 | 0.0001 | 0.001 | Bigfish | 9 |
> | 8 | 8 | 0.0001 | 0.0001 | Starpilot | 5 |
> | 8 | 4 | 0.005 | 0.001 | Starpilot | 5 |
> | 8 | 4 | 0.0001 | 0.001 | Fruitbot | 3 |
>
> We also experimented with a modified version, which we refer to as HOC-PPO, where the architecture was adapted to better leverage PPO’s robustness. While this modification somewhat stabilized the options, the resulting performance was still poor. As this is an unpublished method with limited utility, we chose not to include HOC-PPO as a baseline for this work.
>
> **Tabular comparisons**
>
> > • FraCO is not compared against OC-PPO (or any other hierarchical approaches) in the tabular settings; why not?
> >
>
> Thank you for this suggestion. We agree that comparing FraCOs to other hierarchical approaches in tabular settings would add value. However, our primary goal with the tabular experiments was to validate the conceptual soundness of FraCOs and directly show that increasing the hierarchy depth improves learning efficiency in unseen environments.
>
> While such comparisons in tabular settings are valuable, our main focus was on evaluating FraCOs in deep reinforcement learning environments, where robust and effective multi-level hierarchical methods are scarce. FraCOs’ success in these challenging settings underscores its potential, which we felt was the most impactful demonstration for this work.

---

> > ### Comment · Reviewer_11zq · 2024-11-24
> >
> > I thank the authors for their response. I maintain my positive view of this work, as well as my present score.

---

### Author Response · Authors · 2024-11-19
**Summary of changes.**

We sincerely thank all reviewers for their detailed and constructive feedback. Your insights have significantly enhanced the overall quality of our work. Below, we summarise the key changes made in response to your comments:

- **Statistical reliability**: Procgen experiments have been re-run with **eight seeds** to improve robustness.
- **Fairer comparisons**: All plots now use the **Interquartile Mean (IQM)** as recommended.
- **Hyperparameter search**: A thorough search was conducted for OC-PPO, and are included in Appendix A.13.
- **Ablation studies**:
    - The **Expected Usefulness weightings** have been analysed, with results added to Appendix A.17.
    - Additional insights into clustering approaches, including HDBSCAN hyperparameter sweeps, are now provided in Appendix A.16.
- **Expanded results**:
    - Full results across **all tested Procgen environments** are now presented in the main text, with learning curves detailed in Appendix A.14.
    - Results for **PPG** have been added for completeness.
    - Full FraCOs and FraCOs-SSR results are included in the main body.
- **Improved clarity**:
    - A **glossary of terms** has been added at the start of the appendix to aid navigation.

We hope these changes address the reviewers' concerns and improve the clarity and rigour of our work. We sincerely appreciate your thoughtful feedback and the time you’ve dedicated to reviewing our submission. Your insights have been invaluable in refining this research, and we look forward to any further discussions or suggestions you may have. Thank you for considering our work.

---

### Meta-Review · Area_Chair_XBVf · 2024-12-20

**Metareview:**

This paper presents a method for constructing options via clustering "fractures", which are defined as a state and a finite-length sequence of actions. The authors present a method for measuring the expected usefulness of the discovered fractures and select the ones with highest expected usefulness.

The paper is clear and well-written, the method novel, and empirical evaluations demonstrate its strength.

While there were some concerns on the baselines used, in particular with the tabular experiments, I do agree with the authors that the tabular settings are mostly meant as a validation of the method, rather than as a proper empirical comparison against other methods (which were developed for deep RL environments).

There were also some concerns raised as to whether the claims of "accelerating generalization" are justified, especially given the strong title. It seems this claim is mostly evaluated by running on the ProcGen suite, rather than through a more systematic study (beyond the tabular visualizations). I tend to agree with this sentiment and would suggest the authors temper the claims of "increased generalization".

Despite these drawbacks, I do feel the paper presents an interesting and novel algorithm that may be useful for others in the community. I would strongly suggest the authors incorporate the feedback provided by the reviewers, in particular with respect to the strenghts of the claims.

**Additional Comments On Reviewer Discussion:**

There were quite a few requests on strengthening the empirical evaluations.  The authors significantly improved on this front by running more seeds, conducting a hyper-parameter sweep on one of the baselines, and reporting more statistically significant metrics.

---

### Decision · Program_Chairs · 2025-01-22

Accept (Poster)